# Constrained Policy Optimization with Explicit Behavior Density for Offline Reinforcement Learning

**Jing Zhang**
HKUST
jzhanggy@connect.ust.hk

**Chi Zhang**
Kuaishou Technologies
zhangchi08@kuaishou.com

**Wenjia Wang**[*]
HKUST (GZ) and HKUST
wenjiawang@ust.hk

**Bing-Yi Jing**[*]
SUSTech
jingby@sustech.edu.cn

## Abstract

Due to the inability to interact with the environment, offline reinforcement learning (RL) methods face the challenge of estimating the Out-of-Distribution (OOD) points. Existing methods for addressing this issue either control policy to exclude the OOD action or make the $Q$ function pessimistic. However, these methods can be overly conservative or fail to identify OOD areas accurately. To overcome this problem, we propose a Constrained Policy optimization with Explicit Behavior density (CPED) method that utilizes a flow-GAN model to explicitly estimate the density of behavior policy. By estimating the explicit density, CPED can accurately identify the safe region and enable optimization within the region, resulting in less conservative learning policies. We further provide theoretical results for both the flow-GAN estimator and performance guarantee for CPED by showing that CPED can find the optimal $Q$-function value. Empirically, CPED outperforms existing alternatives on various standard offline reinforcement learning tasks, yielding higher expected returns.

## 1 Introduction

As a form of active learning, reinforcement learning (RL) has achieved great empirical success in both simulation tasks and industrial applications [1, 2, 3, 4, 5, 6, 7]. The great success of RL is largely due to sufficient information exploration and the online training paradigm that the agent has plenty of interactions with the environment. However, the merits of RL methods are limited when it is difficult or costly to sequentially interact with the environment. In many real-world scenarios such as driverless or intelligent diagnostics [8], online interaction is unacceptable. Thus, vanilla RL methods may possibly fail under such circumstances.

One recently developed approach to mitigating the constrained online-interaction is offline RL [8, 9, 10]. Motivated by various data-driven techniques, the offline RL leverages prior experiences to train a policy without interacting with the environment. In offline RL, a critical challenge is distribution shift (also called "extrapolation error" in literature). Specifically, vanilla RL methods

---

[*]Corresponding authors: Bing-Yi Jing and Wenjia Wang.

37th Conference on Neural Information Processing Systems (NeurIPS 2023).

using the Bellman equation and the $Q$-function approximation probably fail to deliver good estimates for Out-of-Distribution (OOD) points. To this end, two types of offline RL solutions have been proposed. One type is the $Q$-function constraint. By adding constraints on the $Q$-function, this type of solution provides a pessimistic estimation of the $Q$-function and prevents the $Q$-value from being too large. The constraints include ensembling multiple $Q$-functions [11, 12], penalizing $Q$-function with high uncertainty [13, 14], among others. Recently, implicit Q-learning [15] and mildly conservative Q-learning [16] are suggested. However, the uncertainty of the $Q$-function or a tight lower bound of $Q$-value is difficult to estimate. Thus the performance is not guaranteed neither theoretically nor practically.

Another type of solution is policy control, where the learning policy is controlled over the support of the behavior policy. Thus the OOD points will not be visited and the distribution shift problem is alleviated [17, 18, 19]. It is noted that previous policy control methods limit the safe areas within the close neighborhoods of the training dataset by distribution distance metrics such as KL divergence, maximum mean discrepancy (MMD), and Wasserstein Distance [20, 17, 18]. Such policy control approaches often lead to an over-conservative and suboptimal learning policy, and the optimal policy value may not be reached. For example, if some state-action points are within the training distribution but unobserved in the training dataset, they may be considered OOD points in previous policy control approaches. Nevertheless, these points are safe when the estimated $Q$-function can well generalize on this area [12, 21], and the optimal policy likely lies in the unseen area of the data space. In this sense, a global estimator of the conditional action data space is required to define the safe areas and examine whether an unseen point is safe. To this end, Wu et al. [22] imposes the density-based constraints and proposes the SPOT method. However, the density estimator in SPOT approximates the lower bound of the behavior policy density, and policy control methods with more accurate density estimators are worth further discussing. As in the field of policy control methods for offline RL problem, the idea of ensuring the consistency of the support of learned policy and that of the behavior policy is considered the most desirable approach to tackle distribution shift. Therefore, a direct estimation density of the behavior policy is desired for guarantee support consistency.

Recently, generative models have shown strong capabilities in diverse fields, in which variational autoencoder (VAE) and generative adversarial network (GAN) [23, 24, 25, 26] are popular tools utilized for sampling and other related applications. In RL areas, both VAE and GAN have been introduced in RL methods [19, 16, 27, 28]. However, as random sample generators, VAE and GAN cannot be directly applied to estimate behavior densities in offline RL, nor can they provide accurate density estimates. Wang et al. [29], Yang et al. [30], Chen et al. [31], Ghasemipour et al. [32] have shown that VAE has shortcomings in estimating complex distributions, especially multimodal distributions or complex behavior policy distributions, and discuss that typically-used VAEs may not align well with the behavior dataset. As VAE is prone to cover mode, this phenomenon diminishes the VAE's capacity to accurately estimate data distributions, particularly in scenarios involving multimodal data distributions, such as behavior policy distributions. Compared to VAE, GAN has more advantages in learning complex data distributions. Recently, various GAN variants have been suggested. Among them, the Flow-GAN [33] model combines the normalizing flow model [34, 35] and GAN, and provides an explicit density estimator. The Flow-GAN model is further renowned for its robustness and tuning flexibility and has garnered increasing attention.

In this work, we propose a novel approach called Constrained Policy optimization with Explicit behavior Density (CPED[2]) for offline RL. The CPED suggests a strong density-based constraint and employs an explicit and accurate density estimator to identify the safe areas. Specifically, CPED utilizes the Flow-GAN model as the density estimator, whose accuracy is theoretically guaranteed. By leveraging this capability, CPED can accurately identify the feasible region, which includes both observed and unobserved but safe points (illustrated in Figure 1 c). By expanding the feasible region and enabling reasonable exploration within it, CPED exhibits reduced risk-aversion and demonstrates an enhanced ability to generate superior policies.

To summarize, the merits of the proposed CPED include the following.

- By introducing the Flow-GAN model, the CPED directly tackles the density estimation problem and proposes a novel policy control scheme for solving offline RL tasks.

---

[2]Code available at https://github.com/evalarzj/cped

- The CPED does not behave over-conservatively and allows exploration in the safe region, especially high-density areas, to find the optimal policy.

- Practically, the effectiveness of CPED is fully demonstrated empirically. Extensive experiments on competitive tasks show that CPED substantially outperforms state-of-the-art methods.

- From the theoretical perspective, we show that GAN with a hybrid loss can accurately estimate density, which itself contributes to the GAN community. Furthermore, we prove the policy learned from CPED can access the optimal value function, theoretically ensuring the efficiency of CPED.

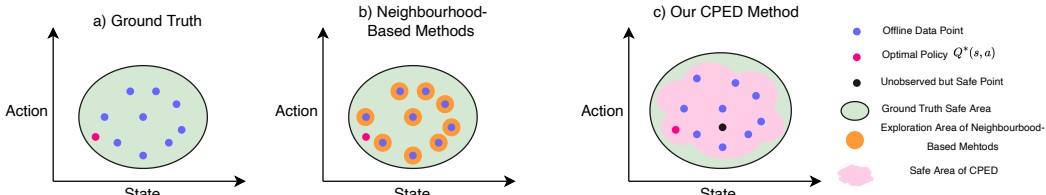

Figure 1: (a): The ground truth safe area in offline RL optimization, and the updates of policies and $Q$-functions are done within the green area. The blue points are collected behavior data $\mathcal{D}$, and the red point denotes the optimal policy given the states. (b): In previous approaches, the exploration of the policy takes place in a small neighborhood of points in $\mathcal{D}$ (the orange circles). (c): The CPED relaxes the exploration area and constructs the feasible region (pink areas), which includes the unobserved but safe points (black point).

## 2 Related Works

**Value Function Constraint**. Taking OOD actions can lead to overestimated value function and introduce bootstrap error. Value function constraint methods aim to construct pessimistic value functions or reduce the uncertainty of the value function to avoid OOD actions. Following [11, 12, 36], a pessimistic value function can be obtained by ensembling multiple value functions. CQL [13] proposes a regularization term so that the estimated $Q$-function low-bounds its true value. TQC [37] follows the distributional RL method and learns the percentile of value function distribution. IQL [15] combines expectile regression with exponential weighted advantage function to avoid extreme value function. MCQ [16] tries to use a special bellman operator to separate the estimation of the target value function for the in-distribution point and OOD point. The value function shows great uncertainty when taking OOD points. UWAC [14] estimates the uncertainty of the value function using the dropout technique. Urpí et al. [38] applies the tail risk measurement for the $Q$-function, in which the value function with high uncertainty is penalized, and the learned value function is forced to be pessimistic. However, finding the lower bound of the value function or estimating the uncertainty is not easy. The error from estimating the value function and finding the lower bound hinders the performance of value function constraint methods. Recently, a series of in-sample learning methods [39, 40, 41, 42, 43] have been proposed to optimize the $Q$-value function or the policy value function $V$ solely utilizing in-sample data. As a result, the extrapolation error, induced by out-of-distribution (OOD) data points in the $Q$-value function, can be mitigated.

**Policy Control Method**. To prevent the learning of policies that take out-of-distribution (OOD) actions, the policy control method incorporates a penalty term during policy learning and restricts the distribution distance between the learned policy and the behavior policy. The BEAR method [17] uses the MMD distance to constrain the difference between the behavior policy and the learned policy. [18, 44, 20, 45, 46] measures the distribution shift between learning policy and behavior policy with KL divergence and Wasserstein distance. DOGE [46] learns a distance function to control the learning policy staying close to the behavior policy. Besides the explicit distance-based policy control methods, some implicit control methods [19, 47, 22], which generate the learning policy action based on the estimated conditional probability distributions, are suggested. However, explicit distance control methods suffer from the generalization problem since the distance is estimated from the limited training samples. On the other hand, implicit control methods tend to generate over-conservative

policies, where the generated actions are too close to the behavior policy. Consequently, the learned policy by these methods discourages exploration and remains conservative.

**Behavior Policy Density Estimation** In offline RL setting, determining the action space $\mathscr{A}$ under given states is crucial. This is equivalent to learning the distribution of the behavior policy. However, traditional density estimation methods [48, 49, 50, 51, 52, 53] often fail in this scenario of density estimation due to the curse of high dimensionality. Recently, deep neural network-based density estimation methods have proven successful in learning the distribution of the behavior policy. Conditional variational autoencoders (CVAE) have been utilized as both behavior policy samplers [19, 17, 47, 16, 54] and density estimators [22]. Generative adversarial networks (GAN) are also introduced in offline RL scenario [55, 56]. Singh et al. [57] applies the generative normalized flow model to learn an invertible map from the noisy prior to the behavior policy action space and do behavior policy sampling by the trained model. EMaQ [32] uses the Autoregressive Generative Model instead of CVAE as a sampler. Additionally, the diffusion model has been applied to estimate the density of the behavior policy in both model-free RL [31] and model-based RL [58]. Specifically, SPOT [22] approaches the lower bound of the behavior density with ELBO derived from CVAE. However, to the best of our knowledge, most of the aforementioned methods either estimate the behavior density implicitly or lack the ability to provide accurate estimations. In our proposed method, the CPED finds an exact density function using the Flow model [34, 35] and GAN [23, 24, 25, 59, 60].

## 3 Probability Controlled Offline RL Framework

### 3.1 Offline RL Problem Settings

In RL setting, the dynamic system is described in terms of a Markov decision process (MDP), which can be defined by a set of tuples $\{\mathcal{S}, \mathcal{A}, T, p_0, r, \gamma\}$. Here, $\mathcal{S}$ and $\mathcal{A}$ denote the state space and action space, respectively; $T$ is a conditional transition kernel, and $T(s'|s, a)$ is the conditional transition probability between states, describing the dynamics of the entire system; $p_0$ is the distribution of the initial states; $r(s, a)$ is a deterministic reward function; and $\gamma \in (0, 1)$ is a scalar discount factor.

The goal of RL is to find a distribution of actions $\pi(a|s)$ (named as policy) that yields the highest returns within a trajectory, i.e., maximizing the expected cumulative discounted returns. We denote this expected discounted return starting from $(s, a)$ under policy $\pi$ as $Q^\pi(s, a)$. Under the Q-learning framework, the optimal expected discounted return $Q^*(s, a)$ can be obtained by minimizing the $L_2$ norm of Bellman residuals:

$$\mathbb{E}_{s' \sim T(s'|s,a), a' \sim \pi(a|s)}[Q(s, a) - (\mathcal{B}Q)(s, a)]^2, \tag{1}$$

where $\mathcal{B}$ is the Bellman operator defined as

$$(\mathcal{B}Q)(s, a) := r(s, a) + \gamma \mathbb{E}_{s'' \sim T(s'|s,a)}[\max_{a'} Q(s', a')].$$

In the context of deep RL, various solutions are provided to estimate the $Q$-function [61, 62, 63]. When the action space is continuous, the learning process is divided into actor ($Q$ value function) training and critic (learning policy) training [64]. The critic network is learned from Eq.1 without "maximum" in the Bellman operator. The actor is formulated with various forms [65, 66, 67, 1] such as maximizing the $Q$-value (DDPG), the weighted log of action density (policy gradient), and clipped weighted action density ratio (PPO).

In the offline RL, the learning policy cannot interact with the dynamic system yet only learns from a training dataset $\mathcal{D}(s, a, s', r)$. Dataset $\mathcal{D}$ contains many triples $(s, a, s', r)$ that can be viewed as independent. Assume that the dataset $\mathcal{D}$ is generated by a behavior policy $\pi_\beta$. We need to redefine the MDP corresponding to $\mathcal{D}$ under the offline RL settings.

**Definition 3.1** (Offline MDP). *Given a dataset $\mathcal{D}$ with continuous state and action space. Let $\pi_\beta$ denote the behavior policy that generated $\mathcal{D}$. The MDP of dataset $\mathcal{D}$ is defined by $\mathcal{M}_\mathcal{D} := \{\mathscr{S}, \mathscr{A}, T_\mathcal{D}, p_{0_\mathcal{D}}, r, \gamma\}$, where $\mathscr{S} = \{\mathcal{S}, \mathbb{P}_\mathcal{D}\}$ is the measured state space related to $\mathcal{D}$ with probability measure $\mathbb{P}_\mathcal{D}$, $\mathbb{P}_\mathcal{D}$ is the marginal probability of each state in state space of $\mathcal{D}$, $\mathscr{A} = \{\mathcal{A}, \pi_\beta\}$ is the measured action space related to $\mathcal{D}$ with conditional probability measure $\pi_\beta(a|s)$, $T_\mathcal{D}(s'|s, a)$ is the transition probability between states, describing the dynamics of $\mathcal{D}$, and $p_{0_\mathcal{D}}$ is the distribution of the initial states of $\mathcal{D}$.*

In offline RL, we can obtain the optimal $Q$-function by minimizing Eq.1. However, due to the distribution shift problem [17, 8], the Bellman residuals are biased during the training phase. Specifically, at $k$-th iteration, since $\pi_k$ is trained to maximize the $Q$-function, it is possible that $\pi_k$ visits some OOD points with high estimated $Q$-function values. Nevertheless, we cannot evaluate the rewards of the OOD points under offline scenarios. Consequently the Bellman residuals at the OOD points may dominate the loss, and the learned $Q$-function value is mislead. Therefore, the optimization of offline RL must occur in the bounded safe state-action space $\mathscr{S} \times \mathscr{A}$, such that the estimated $Q$-function is trustworthy.

## 3.2 Behavior Policy Estimation by GAN with Specific Density

In offline RL, the dataset $\mathcal{D}$ is generated by the behavior policy, whose density can be estimated via inverse RL [68]. One widely used approach in inverse RL, maximum entropy inverse RL (MaxEnt IRL, [69]), models the density via a Boltzmann distribution

$$p_\theta(\tau) = Z^{-1} \exp(-c_\theta(\tau)), \tag{2}$$

where the energy is given by the cost function $c_\theta(\tau) = \sum_{t=0}^{H} c_\theta(s_t, a_t)$, $\tau = (s_0, a_0, s_1, a_1, ..., s_H, a_H)$ is a trajectory, and $Z$ is a scale factor. The cost function $c_\theta(\tau) = \sum_{t=0}^{H} c_\theta(s_t, a_t)$ is a parameterized cost function that can be learned by maximizing the likelihood of the trajectories in $\mathcal{D}$. And the scale factor $Z$ is the sum of exponential cost $\exp(-c_\theta(\tau))$ of all the trajectories that belong to the product space of the state and action of $\mathcal{D}$. However, estimating $Z$ is difficult because it is scarcely possible to traverse the entire action space $\mathscr{A}$. An alternative way to estimate $Z$ is via guided cost learning [70], where it has been shown that learning $p_\theta(\tau)$ is equivalent to training the dataset $\mathcal{D}$ with a GAN [70, 71, 72, 73].

Originally, the GAN is a random sample generator for high-dimensional data, and it consists of two models: a generator $G$ and a discriminator $D$. The generator is trained to generate random samples that are close to those drawn from the underlying true distribution, and the discriminator tries to classify whether the input is generated sample or an actual one from the training dataset. The GAN cannot provide an explicit density estimation, which is an indispensable ingredient when controlling the probability of actions and specifying an safe area. Thus, we cannot directly apply the MaxEnt IRL to our CPED. To overcome the difficulty, we follow the approach in [33] and consider GAN with flow model as a generator (we call if Flow-GAN hereafter as in [33]).

Flow-based generative models have shown their power in the fields of image processing and natural language processing [34, 35]. The normalizing flow model allows us to start from a simple prior noise distribution and obtain an *explicit* estimate of the density function after a series of non-linear invertible transformations.

Specifically, let $f_\theta : \mathbb{R}^A \mapsto \mathbb{R}^A$ be a nonlinear invertible transformation, where $A$ is the dimension of action space $\mathscr{A}$, and $\theta$ is parameter. By change of variables, the probability density on $\mathcal{D}$ can be obtained by

$$p_\theta(\tau) = p_H(f(\tau)) \left| \det \frac{\partial f_\theta(\tau)}{\partial \tau} \right|, \tag{3}$$

where $\tau \in \mathcal{D}$ is a trajectory. Since $f_\theta$ is invertible, a generator can be constructed by $G_\theta = f_\theta^{-1}$. In practice, both generator and discriminator in GAN are modeled as deep neural networks. Two specific neural network structures, NICE [34] and Real_NVP [35], are proposed to ensure that $G_\theta$ (or $f_\theta$) is invertible. The Jacobian matrices of the mapping functions $f$ constructed by these two methods are both triangular matrices, guaranteeing efficient computation of $|\det \frac{\partial f(\tau)}{\partial \tau}|$.

Directly training a Flow-GAN by original loss function in GAN may lead to poor log-likelihoods, and GAN is prone to mode collapse and producing less diverse distributions [33]. Thus, following the approach in [33], we adopt a hybrid loss function containing a maximum likelihood loss and a GAN structure loss as the optimization objective for estimating the probability measure of behavior policy:

$$\min_\theta \max_\phi \mathcal{L}_\mathcal{G}(G_\theta, D_\phi) - \lambda \mathbb{E}_{\tau \sim P_{data}}[\log(p_\theta(\tau))] \tag{4}$$

where $\mathcal{L}_\mathcal{G}(G_\theta, D_\phi)$ can be any min-max loss function of GAN, $D_\phi$ is a discriminator indexed by parameter $\phi$, and $\lambda > 0$ is a hyperparameter balancing the adversary learning process and the maximum likelihood process.

*Remark* 3.1. The reason for not simply using MLE is that MLE tends to cover all the modes of distribution (even though some modes have small probabilities), and MLE is not robust against model misspecification [74], which often occurs in high dimensional scenarios.

In the following proposition, we show that the estimated density of the behavior policy using the hybrid loss Eq.4 is equivalent to that using MaxEnt IRL. The proof of Proposition 3.1 is provided in Appendix B.

**Proposition 3.1.** *For offline dataset $\mathcal{D}$ generated by behavior policy $\pi_\beta$, the learned likelihood function $L^{\pi_\beta}$, using GAN with hybrid loss in Eq.4 is equivalent to that trained by MaxEnt IRL. If the generator of GAN can give a specific likelihood function $p_\theta^G(\tau)$, then*

$$L_\theta^{\pi_\beta}(\tau) = CZ^{-1}\exp(-c_\theta(\tau)) \propto p_\theta^G(\tau) \tag{5}$$

*where $C$ is a constant related to $\mathcal{D}$.*

### 3.3 Constrained Policy Optimization with Explicit Behavior Density Algorithm (CPED)

In this work, we propose an algorithm for probabilistically controlling the learned policy, CPED. In CPED, we first utilize the Flow-GAN and obtain a probability estimate of the trajectory $p_\theta(\tau)$. The optimize objective follows the hybrid loss in Eq.4:

$$\min_\theta \max_\phi \mathcal{L}_\mathcal{G}(G_\theta, D_\phi) - \lambda \mathbb{E}_{(s,a)\sim D}[\log(L_\theta^{\pi_\beta}(s,a)))]. \tag{6}$$

Since the episode $\tau$ is a sequential collection of states $s$ and actions $a$, we rewrite the input of $L_\theta^{\pi_\beta}(\cdot)$ with $(s,a)$ instead of $\tau$ in Eq.6 and afterwards. From Proposition 3.1, the solution to Eq.6 is equivalent to the behavior estimator obtained from MaxEnt IRL, thus providing an effective estimation of the behavior policy.

Following the learning paradigm in classic RL methods, the policy learning process is divided into two parts: the actor training and critic training. With the estimated function $L_\theta^{\pi_\beta}(\cdot)$ approximating the density of behavior policy, we can naturally propose a density-based constraint in policy learning, and obtain the safe area according to the estimated density. The points with low density values are excluded and the updated policy cannot are exceed the safe region. The actor learning is formulated as:

$$\max_\psi \mathbb{E}_{s\sim\mathbb{P}_\mathcal{D}, a\sim\pi_\psi(\cdot|s),(s,a)\in\tilde{\mathcal{S}}\times\tilde{\mathcal{A}}}[Q_\eta(s,a)], \tag{7}$$

where $\pi_\psi(\cdot|s)$ is the learning policy, and $\tilde{\mathcal{S}} \times \tilde{\mathcal{A}}$ is an estimate of the bounded safe area $\mathscr{S} \times \mathscr{A}$ in Definition 3.1.

There are many approaches for specifying the safe area $\tilde{\mathcal{S}} \times \tilde{\mathcal{A}}$. Kumar et al. [17] bounds the expected MMD distance between the learning policy and the behavior policy within a predetermined threshold, while Wu et al. [22] constrains the lower bound of the estimated behavior density to identify the safe region. In this work, since we can have a direct density estimation of $L_\theta^{\pi_\beta}(s,a)$, we can utilize it and specify the safe region as the region with not too small density. Specifically, we set $\tilde{\mathcal{S}} \times \tilde{\mathcal{A}} = \{(s,a) \in \mathcal{S} \times \mathcal{A} : -\log L_\theta^{\pi_\beta}(s,a) < \epsilon\}$, where $\epsilon$ is the threshold[3].

The critic training is following the traditional Bellman equation as shown in Eq.1. The entire algorithm of CPED is summarized in Algorithm 1. In CPED, the Flow-GAN model plays a crucial role in estimating the behavior density. The density function is obtained from the generator model and is iteratively updated along with the discriminator. We further use the TD3 framework to solve the RL problem. It is noted that during Actor updates, the density of sampled actions should be higher than the threshold to avoid out-of-distribution areas.

## 4 Theoretical Analysis

In this section, we provide theoretical analysis of our proposed method. In Section 4.1, we prove the convergence of GAN with the hybrid loss, and based on this result, we show the convergence of CPED in Section 4.2.

---

[3]Eq.7 is a constrained optimization problem, and we are indeed using Lagrangian techniques to solve the constraint problem. During the practical optimization, the constraint on $\tilde{\mathcal{S}} \times \tilde{\mathcal{A}} = \{(s,a) \in \mathcal{S} \times \mathcal{A} : -\log L_\theta^{\pi_\beta}(s,a) < \epsilon\}$ in Eq.7 works as penalty terms in the Lagrange function (with Lagrangian multiplier $\alpha$) .

**Algorithm 1** The CPED algorithm

---

**Input:** dataset $\mathcal{D}$, target network update rate $\kappa$, Lagrange multiplier $\alpha$, training ratio of the generator and discrimination $r$, mini-batch size $N$, hyperparameters $\lambda, c$.

Initialize generator $G_\theta$, discriminator $D_\phi$, loss function in GAN $\mathcal{L}_\mathcal{G}(\cdot)$, behavior policy likelihood $L_\theta^{\pi_\beta}(\cdot)$, Q networks $\{Q_{\eta_1}, Q_{\eta_2}\}$, actor $\pi_\psi$, target networks $\{Q_{\eta_1'}, Q_{\eta_2'}\}$, target actor $\pi_{\psi'}$, with $\psi' \leftarrow \psi, \eta_i' \leftarrow \eta_i, i = 1, 2$.

**for** $i = 1$ **to** $M$ **do**
    Sample mini-batch of transitions $(s, a, r, s') \sim \mathcal{D}$
    **Initial Training Flow-GAN:**
    $\phi \leftarrow \mathrm{argmax}_\phi \mathcal{L}_\mathcal{G}(G_\theta, D_\phi)$
    $\theta \leftarrow \mathrm{argmin}_\theta \max_\phi \mathcal{L}_\mathcal{G}(G_\theta, D_\phi) - [\lambda \mathbb{E}_{(s,a)\sim\mathcal{D}} \log(L_\theta^{\pi_\beta}(s, a))]$
**end for**
**for** $i = 1$ **to** $N$ **do**
    Sample mini-batch of transitions $(s, a, r, s') \sim \mathcal{D}$
    **Updating Flow-GAN:**
    $\phi \leftarrow \mathrm{argmax}_\phi \mathcal{L}_\mathcal{G}(G_\theta, D_\phi)$
    $\theta \leftarrow \mathrm{argmin}_\theta \max_\phi \mathcal{L}_\mathcal{G}(G_\theta, D_\phi) - [\lambda \mathbb{E}_{(s,a)\sim\mathcal{D}} \log(L_\theta^{\pi_\beta}(s, a))]$
    **Updating Q-function:**
    Get action for the next state, $\{\tilde{a} \sim \pi_{\psi'}(\cdot|s') + \varepsilon, \varepsilon \sim clip(\mathcal{N}(0, \sigma), -c, c)\}$
    Let $y(s, a) := \min(Q_{\eta_1'}(s', \tilde{a}), Q_{\eta_2'}(s', \tilde{a}))$
    $\eta_i \leftarrow \mathrm{argmin}_{\eta_i}(Q_{\eta_i}(s, a) - (r + \gamma y(s, a)))^2, i = 1, 2$
    **Updating Actor:**
    Update $\psi$ according to Eq.7, by using dual gradient descent with Lagrange multiplier $\alpha$
    **Update Target Networks:**
    $\psi' \leftarrow \kappa\psi + (1 - \kappa)\psi'; \eta_i' \leftarrow \kappa\eta_i + (1 - \kappa)\eta_i', i = 1, 2$
**end for**

---

## 4.1 Convergence of GAN with the Hybrid Loss

In this subsection, we show that training GAN with the hybrid loss function Eq.4 can yield an accurate density estimator. For ease of presentation, assume we observed $X_j \sim p^* = \frac{\mathrm{d}\mu^*}{\mathrm{d}\nu}, j = 1, ..., n$, where $\mu^*$ is the underlying true probability measure (the probability measure of behavior policy), and $\nu$ is the Lebesgue measure. Motivated by [24, 75, 76], we consider the Integral Probability Metric (IPM) defined as

$$d_{\mathcal{F}_d}(\mu_1, \mu_2) := \sup_{f \in \mathcal{F}_d} \mathbb{E}_{X \sim \mu_1} f(X) - \mathbb{E}_{Y \sim \mu_2} f(Y), \tag{8}$$

where $\mu_1$ and $\mu_2$ are two probability measures, and $\mathcal{F}_d$ is a discriminator class. Under IPM, the GAN framework with the hybrid loss Eq.4 solves the empirical problem (cf., Eq. (1.1) of [75] for the original GAN loss function)

$$\mu_n \in \underset{\mu \in \mathcal{Q}_g}{\mathrm{argmin}} \max_{f \in \mathcal{F}_d} \int_\Omega f \mathrm{d}\mu - \int_\Omega f \mathrm{d}\tilde{\mu}_n - \lambda \int_\Omega \log p \mathrm{d}\hat{\mu}_n, \tag{9}$$

where $\hat{\mu}_n = \frac{1}{n} \sum_{j=1}^n \delta_{X_j}$ is the empirical measure, $\tilde{\mu}_n$ is a (regularized) density estimation, $\mathcal{Q}_g$ is the generator class, and $p = \frac{\mathrm{d}\mu}{\mathrm{d}\nu}$. In the following (informal) theorem, we show that $\mu_n$ is an accurate estimator of the underlying true distribution. A detailed version of Theorem 4.1 and its proof can be found in Appendix A.

**Theorem 4.1** (Informal). *Suppose the generator class $\mathcal{Q}_g$ and the discriminator class $\mathcal{F}_d$ are induced by some Sobolev spaces. Choose $\lambda$ as a constant. Under certain conditions,*

$$d_{\mathcal{F}_d}(\mu^*, \mu_n) = O_\mathbb{P}(n^{-1/2}), \mathrm{KL}(\mu^*||\mu_n) = O_\mathbb{P}(n^{-1/2}),$$

*where $\mu^*$ is the true probability measure, $\mu_n$ is as in Eq.9 with hybrid loss, and $\mathrm{KL}(\mu^*||\mu_n)$ is the KL divergence.*

Theorem 4.1 states a fast convergence rate $O_\mathbb{P}(n^{-1/2})$ of the IPM and KL divergence can be achieved. Therefore, applying the hybrid loss can provide not only a good generator, but also provide an accurate density estimate, which can be utilized in our CPED framework to control the safe region.

*Remark* 4.1. Given the universal approximation theorem [77, 78], the neural networks can approximate any smooth function. Thus, we consider Sobolev function classes instead of neural networks for the ease of mathematical treatment. If both the generator and discriminator classes are neural networks, one may adopt similar approach as in [75, 79] and then apply Theorem 4.1.

*Remark* 4.2. In [75], the convergence rate of $\mathbb{E}d_{\mathcal{F}_d}(\mu^*, \mu_n)$ is considered, while we provide a sharper characterization of $d_{\mathcal{F}_d}(\mu^*, \mu_n)$, by showing the convergence is in probability with certain convergence rate.

## 4.2 Convergence of CPED

With an accurate estimate of the underlying density, in this subsection, we give a theoretical analysis of how CPED can find the optimal $Q$-function value over the product space of $\mathscr{S} \times \mathscr{A}$. The proof of Theorem 4.2 can be found in Appendix C.

**Theorem 4.2.** *Suppose $Q^*(s, a)$ is the optimal $Q$-value on product space $\mathscr{S} \times \mathscr{A}$. Let $\hat{\pi}(\cdot|s) := \arg\max_a Q^*(s, a)$ be a learning policy. Let $\hat{\pi}^\Delta(\cdot|s)$ be the probability-controlled learning policy using CPED, which is defined on the support of $\pi_\beta$. Suppose $\hat{\pi}^\Delta(\cdot|s) > 0$ on $\mathscr{A}$. Then with probability tending to one,*

$$Q^*(s, a) = Q(s, \hat{\pi}^\Delta(\cdot|s)). \tag{10}$$

Furthermore, when minimizing the Bellman residual to solve the optimal $Q$-function and training the policy by iteration, we have the following theorem, whose proof is provided in Appendix D.

**Theorem 4.3.** *Suppose in step $k + 1$, the learned policy is defined as $\pi_{k+1} := \arg\max_a Q^{\pi_k}(s, a), \forall s \in \mathscr{S}$. Let $V^{\pi_k}$ be the value function related to $\pi_k$ in step $k$, defined as $V^{\pi_k}(s) = \mathbb{E}_{a \sim \pi_k(\cdot|s)}[Q(s, a)]$, and $V^*$ be the optimal value function over the product space $\mathscr{S} \times \mathscr{A}$. Let $\hat{\pi}_k^\Delta$ be the probability-controlled learning policy using CPED in step $k$, which is defined on the support of $\pi_\beta$. Then with probability tending to one,*

$$\|V^{\hat{\pi}_{k+1}^\Delta} - V^*\|_\infty \leq \gamma^{k+1}\|V^{\hat{\pi}_0^\Delta} - V^*\|_\infty. \tag{11}$$

Theorem 4.3 provides assurance that CPED can achieve the optimal value function with a linear convergence rate as the iteration number approaches infinity. Despite under offline settings, this rate is consistent with the standard rate in online RL [80]. When combined with Theorems 4.1 and 4.2, we can confidently conclude that CPED's effectiveness is theoretically guaranteed.

## 5 Experiments

In the experimental section, we analyze the performance of CPED on several standard offline RL tasks [81]. We compare CPED with a wide range of offline RL methods and analyze the performance of CPED. Further implementation details are provided in Appendix E.2.

### 5.1 Performance Comparison on Standard Benchmarking Datasets for Offline RL

The proposed CPED algorithm is evaluated on the D4RL [81] Gym-MuJoCo and AntMaze tasks. Details of these tasks are shown in Appendix E.1. As a model-free method, CPED is compared with the following model-free baselines: behavioral cloning (BC) method, BCQ [19], DT [82], AWAC [44], BEAR [17], Onestep RL [83],TD3+BC [45], CQL [13], IQL [15], SPOT [22], and PLAS [47]. In the experiments, the performance of all alternatives are measured by the average normalized scores in an episode. The experimental results reported in this paper are either from the authors' original experiments or from our replications.

Tables 1 and 2 show the experimental results of Gym-MuJoCo and AntMaze tasks, respectively. In Gym-MuJoCo tasks, CPED outperforms the competitors in most cases. In particular, our method has achieved remarkable success on the halfcheetah and walker2d tasks, especially under the medium-quality dataset. For the medium-expert task, CPED performs slightly inferior to other tasks, which is due to the fact that the expert task needs tight control so that the learning policy stays close to the behavior policy. In the AntMaze tasks, CPED performs slightly worse than IQL but outperforms the other baselines, and it achieves better returns under "unmaze" and "medium-play" tasks.

Table 1: The performance of CPED and other competing methods on the various D4RL Gym-MuJoCo tasks. We report the performance of Average normalized scores and standard deviations over 5 random seeds. med = medium, r = replay, e = expert, hc = halfcheetah, wa = walker2d, ho=hopper

| Dataset | BC | AWAC | DT | Onestep | TD3+BC | CQL | IQL | SPOT | CPED(Ours) |
|---|---|---|---|---|---|---|---|---|---|
| hc-med | 42.6 | 43.5 | 42.6 | 48.4 | 48.3 | 44.0 | 47.4 | 58.4 | **61.8**±1.6 |
| ho-med | 52.9 | 57.0 | 67.6 | 59.6 | 59.3 | 58.5 | 66.2 | 86.0 | **100.1**±2.8 |
| wa-med | 75.3 | 72.4 | 74.0 | 81.8 | 83.7 | 72.5 | 78.3 | 86.4 | **90.2**±1.7 |
| hc-med-r | 36.6 | 40.5 | 36.6 | 38.1 | 44.6 | 45.5 | 44.2 | 52.2 | **55.8**±2.9 |
| ho-med-r | 18.1 | 37.2 | 82.7 | 97.5 | 60.9 | 95.0 | 94.7 | **100.2** | 98.1±2.1 |
| wa-med-r | 26.0 | 27.0 | 66.6 | 49.5 | 81.8 | 77.2 | 73.8 | 91.6 | **91.9**±0.9 |
| hc-med-e | 55.2 | 42.8 | 86.8 | **93.4** | 90.7 | 91.6 | 86.7 | 86.9 | 85.4±10.9 |
| ho-med-e | 52.5 | 55.8 | **107.6** | 103.3 | 98.0 | 105.4 | 91.5 | 99.3 | 95.3±13.5 |
| wa-med-e | 107.5 | 74.5 | 108.1 | 113.0 | 110.1 | 108.8 | 109.6 | 112.0 | **113.04**±1.4 |
| Total | 466.7 | 450.7 | 672.6 | 684.6 | 677.4 | 698.5 | 692.4 | 773.0 | **791.7**±37.8 |

Table 2: The performance of CPED and other competing methods on the various AntMaze tasks. We report the performance of Average normalized scores and standard deviations over 5 random seeds. med = medium, d = diverse, p = play.

| Dataset | BCQ | BEAR | BC | DT | TD3+BC | PLAS | CQL | IQL | SPOT | CPED(Ours) |
|---|---|---|---|---|---|---|---|---|---|---|
| umaze | 78.9 | 73.0 | 49.2 | 54.2 | 73.0 | 62.0 | 82.6 | 89.6 | 93.5 | **96.8**±2.6 |
| umaze-d | 55.0 | 61.0 | 41.8 | 41.2 | 47.0 | 45.4 | 10.2 | **65.6** | 40.7 | 55.6±2.2 |
| med-p | 0.0 | 0.0 | 0.4 | 0.0 | 0.0 | 31.4 | 59.0 | 76.4 | 74.7 | **85.1**±3.4 |
| med-d | 0.0 | 8.0 | 0.2 | 0.0 | 0.2 | 20.6 | 46.6 | 72.8 | **79.1** | 72.1±2.9 |
| large-p | 6.7 | 0.0 | 0.0 | 0.0 | 0.0 | 2.2 | 16.4 | **42.0** | 35.3 | 34.9±5.3 |
| large-d | 2.2 | 0.0 | 0.0 | 0.0 | 0.0 | 3.0 | 3.2 | **46.0** | 36.3 | 32.3±7.4 |
| Total | 142.8 | 142.0 | 91.6 | 95.4 | 120.2 | 164.6 | 218.0 | **392.4** | 359.6 | 376.8±23.8 |

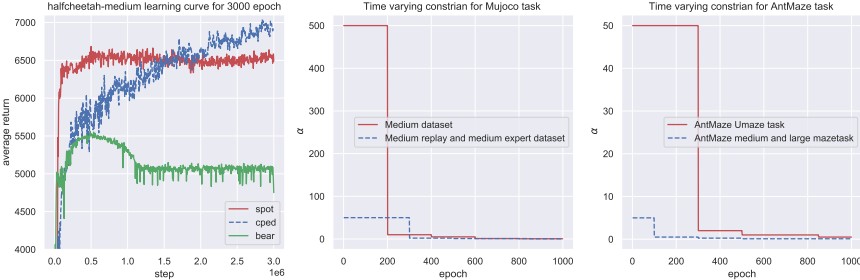

Figure 2: (a) Average performance of BEAR and CPED on halfcheetah-medium task averaged over 5 seeds. BEAR can reach a bottleneck very quickly. CPED remain increasing after reaching the bottleneck. (b) The time(epoch) varying constrain parameter $\alpha$ used in Gym-MuJoCo task. (c) The time(epoch) varying constrain parameter $\alpha$ used in AntMaze task

To further discuss the learning dynamics of the proposed methods, the learning curve of BEAR, SPOT, and CPED in halfcheetah-medium task is shown in Figure 2(a). It is noted that the average return of BEAR increases very fast and remains unchanged after a smooth decrease in subsequent training, indicating that policy exploration is limited. This is consistent with the conservative nature of policy control methods that prevent the learning policy from straying too far from behavior policy. The pattern is similar to SPOT's learning curve, which shows a rapid increase and then enters a plateau. In contrast, the average returns of CPED methods continue to rise after a sharp increase to a

certain extent, indicating that CPED keeps exploring the data and benefits from exploration. This also confirms Theorem 4.3 (see Section C) from the experimental perspective.

## 5.2 Ablation Study

**Time varying hyperparameter $\alpha$ in policy control methods.** When improving actor using Eq.7, a Lagrange hyperparameter $\alpha$ is needed for this constrained optimization problem. The larger the $\alpha$, the more restricted actions taken by the learning policy. Previous methods either use an auto-learning $\alpha$ [17, 14] or a constant $\alpha$ [18, 22]. BRAC [18] find that auto-learning $\alpha$ performs worse than the constant $\alpha$. In our experiment, we apply a time (epoch) varying $\alpha$ scheme with piece-wise constant functions and find this scheme performs better than the constant setting. The rationale behind this time-varying scheme is that we employ strong constraints for the action taken at the early stage of policy training in case the model fails [8]. The constraint is further relaxed by decreasing $\alpha$ when the policy starts to converge. The piece-wise function of $\alpha$ for the Mujoco tasks and AntMaze tasks are plot in Figure 2(b)-(c), and the performance results of the Mujoco tasks are shown in Figure E.5 in in Appendix E.5. We can see that the Mujoco task significantly benefits from this time varying scheme and delivers better returns in most scenarios.

**Likelihood threshold $\epsilon$.** The hyperparameter $\epsilon$ determines the tightness of our estimated safety boundaries. A larger $\epsilon$ leads to more relaxed boundaries and versa vice. A commonly used setting for $\epsilon$ is the quantile of behavior policy density. However, the quantile value is difficult to set when the dimension is large. In our experiment, CPED uses the mean likelihood of the training batch as $\epsilon$, which decreases the effect of the estimation error of the behavior policy. See Appendix E.5 for additional discussion and details.

## 6    Conclusion and Future Work

In this article, we present CPED, a novel approach for policy control in offline RL. CPED leverages the Flow-GAN model to estimate the density of the behavior policy, enabling the identification of safe areas and avoiding the learning of policies that visit OOD points. We validate the effectiveness of CPED both theoretically and experimentally. Theoretical analysis demonstrates that CPED has a high probability of achieving the optimal policy and generating large returns. Experimental results on the Gym-MuJoCo and AntMaze tasks demonstrate that CPED surpasses state-of-the-art competitors, highlighting its superior performance.

While CPED shows great capacity in experiment studies, there are several potential directions for its extension.. Firstly, as a generic method, GAN has diverse variants and it is necessary to try out more powerful GAN structures for density estimation. Secondly, when the feasible region is identified, how to explore the region efficiently remains another important issue, and we leave it to future work. Finally, it is an interesting direction to examine the performance of CPED in more complex scenarios, including dataset generated by multiple behavior policies or multiple agents, etc.

## Acknowledgements

We would like to thank AC and reviewers for their valuable comments on the manuscript. Wenjia Wang was supported by the Guangzhou-HKUST(GZ) Joint Funding Program (No. 2023A03J0019) and Guangzhou Municipal Science and Technology Project (No. 2023A03J0003)

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
