# Appendix

## A Convergence of GAN with the hybrid loss

Before presenting the formal version of Theorem 4.1 and its proof, we introduce some preliminaries. As stated in Theorem 4.1, we assume that both the discriminator class $\mathcal{F}_d$ and the generator class $\mathcal{Q}_g$ are within some Sobolev spaces. We define the Sobolev space via the wavelet bases as in nonparametric statistics [84].

Without loss of generality, let $\Omega = [0,1]^d$. Let $\phi$ be a "farther wavelet" satisfying $r$-regularity (see [85] for details). For $j \in \mathbb{N}$ and $\boldsymbol{s} \in [2^j]^d$, where $[N] = \{1, ..., N\}$, define

$$\phi_{j,\boldsymbol{s}}(\boldsymbol{x}) = \begin{cases} 2^{dj}\phi(2^{dj}\boldsymbol{x} - \boldsymbol{s}), & \text{if } 2^{dj}\boldsymbol{x} - \boldsymbol{s} \in \Omega, \\ 0, & \text{otherwise.} \end{cases}$$

Then, it can be shown that $\{\phi_{j,\boldsymbol{x}}\}_{j \in \mathbb{N}, \boldsymbol{s} \in [2^j]^d}$ forms an orthonormal basis and for each $j \in \mathbb{N}$, if $\boldsymbol{s} \neq \boldsymbol{s}'$, then $\phi_{j,\boldsymbol{s}}(\boldsymbol{x})$ and $\phi_{j,\boldsymbol{s}'}(\boldsymbol{x})$ have disjoint supports [85]. A Sobolev space with smoothness $m$ can be defined as

$$\mathcal{W}^m(\Omega) = \{f \in L_2(\Omega) : \|f\|_{\mathcal{W}^m(\Omega)} < \infty\},$$

where

$$\|f\|^2_{\mathcal{W}^m(\Omega)} = \sum_{j \in \mathbb{N}} 2^{jdm} \left( \sum_{\boldsymbol{s} \in [2^j]^d} \langle f, \phi_{j,\boldsymbol{s}} \rangle^2_{L_2(\Omega)} \right).$$

We further define a ball with radius $R$ in $\mathcal{W}^m(\Omega)$ as

$$\mathcal{W}^m(R) = \{f \in L_2(\Omega) : \|f\|_{\mathcal{W}^m(\Omega)} \leq R\}.$$

Now we are ready to present a formal version of Theorem 4.1 as follows. With an abuse of notation, we write $\mu \in \mathcal{Q}_g$ to denote that the density function induced by $\mu$ is in $\mathcal{Q}_g$. Thus, $\mu \in \mathcal{Q}_g$ is the same as $p = \frac{d\mu}{d\nu} \in \mathcal{Q}_g$, where $\nu$ is the Lebesgue measure. With this notation, we can directly write $\mathcal{Q}_g$ and $\mathcal{F}_d$ are subsets of some Sobolev spaces respectively instead of writing "$\mathcal{Q}_g$ and $\mathcal{F}_d$ are induced by some Sobolev spaces" as in the informal version Theorem 4.1, which simplifies the statement and proof.

**Theorem A.1.** *Suppose the generator class $\mathcal{Q}_g \subset \mathcal{W}^{m_1}(R_1)$ and the discriminator class $\mathcal{F}_d \subset \mathcal{W}^{m_2}(R_2)$ are two Sobolev spaces with $m_1, m_2 > d/2$, and both $\mathcal{Q}_g$ and $\mathcal{F}_d$ are symmetric. Suppose the underlying true density function $p^* \in \mathcal{Q}_g$. Furthermore, assume that $\mathcal{G} := \{g : g = \log(p/p^*)\} \subset \mathcal{W}^{m_1}(R_3)$ for some constant $R_3$. Let $\lambda$ be a constant. Then we have*

$$d_{\mathcal{F}_d}(\mu^*, \mu_n) = O_{\mathbb{P}}(n^{-1/2}), \mathrm{KL}(\mu^*\|\mu_n) = O_{\mathbb{P}}(n^{-1/2})$$

*where $\mu^*$ is the true probability measure, $\mu_n$ is as in Eq.9 with hybrid loss, and $\mathrm{KL}(\mu^*\|\mu_n)$ is the Kullback–Leibler divergence.*

*Remark* A.1. We assume $m_1, m_2 > d/2$ because the Sobolev embedding theorem implies that all functions in the corresponding spaces are continuous.

*Proof of Theorem A.1.* Because $p^* \in \mathcal{Q}_g$ and $\mu_n$ is as in Eq.9, we have

$$d_{\mathcal{F}_d}(\mu_n, \tilde{\mu}_n) - \lambda \int_\Omega \log p_n \mathrm{d}\hat{\mu}_n \leq d_{\mathcal{F}_d}(\mu^*, \tilde{\mu}_n) - \lambda \int_\Omega \log p^* \mathrm{d}\hat{\mu}_n, \tag{A.1}$$

where $p_n = \frac{d\mu_n}{d\nu}$ with $\nu$ the Lebesgue measure. By the basic inequality $\sup_{\boldsymbol{x} \in \Omega}(f(\boldsymbol{x}) + g(\boldsymbol{x})) \leq \sup_{\boldsymbol{x} \in \Omega} f(\boldsymbol{x}) + \sup_{\boldsymbol{x} \in \Omega} g(\boldsymbol{x})$, Eq.A.1 implies

$$d_{\mathcal{F}_d}(\mu_n, \mu^*) - \lambda \int_\Omega \log p_n \mathrm{d}\hat{\mu}_n \leq d_{\mathcal{F}_d}(\mu^*, \tilde{\mu}_n) + d_{\mathcal{F}_d}(\mu_n, \tilde{\mu}_n) - \lambda \int_\Omega \log p_n \mathrm{d}\hat{\mu}_n$$

$$\leq 2d_{\mathcal{F}_d}(\mu^*, \tilde{\mu}_n) - \lambda \int_\Omega \log p^* \mathrm{d}\hat{\mu}_n, \tag{A.2}$$

where we also use the assumption that both $\mathcal{Q}_g$ and $\mathcal{F}_d$ are symmetric in the first inequality.

By

$$d_{\mathcal{F}_d}(\mu_n, \mu^*) \geq 0,$$

Eq.A.2 gives us

$$-\lambda \int_\Omega \log p_n(\boldsymbol{x}) \mathrm{d}\hat{\mu}_n \leq 2 d_{\mathcal{F}_d}(\mu^*, \tilde{\mu}_n) - \lambda \int_\Omega \log p^*(\boldsymbol{x}) \mathrm{d}\hat{\mu}_n. \tag{A.3}$$

Since $\{\phi_{j,s}, j \in \mathbb{N}, s \in [2^j]^d\}$ forms an orthogonal basis in $L_2(\Omega)$, for any functions $f \in W^{m_2}(\Omega)$ and $p \in W^{m_1}(\Omega)$, they have the expansion as

$$f(\boldsymbol{x}) = \sum_{j \in \mathbb{N}} \sum_{s \in [2^j]^d} \langle f, \phi_{j,s} \rangle_{L_2(\Omega)} \phi_{j,s}(\boldsymbol{x}),$$

$$p(\boldsymbol{x}) = \sum_{j \in \mathbb{N}} \sum_{s \in [2^j]^d} \langle p, \phi_{j,s} \rangle_{L_2(\Omega)} \phi_{j,s}(\boldsymbol{x}). \tag{A.4}$$

Let

$$p_{1,n}(\boldsymbol{x}) = \sum_{j=1}^{M} \sum_{s \in [2^j]^d} b_{j,s} \phi_{j,s}(\boldsymbol{x}), \tag{A.5}$$

where $M$ will be determined later, and

$$b_{j,s} = \frac{1}{n} \sum_{k=1}^{n} \phi_{j,s}(X_k).$$

Thus, plugging Eq.A.4 and Eq.A.5 into $d_{\mathcal{F}_d}(\mu^*, \tilde{\mu}_n)$ yields

$$d_{\mathcal{F}_d}(\mu^*, \tilde{\mu}_n)$$

$$= \sup_{f \in \mathcal{B}_{W^{m_2}(\Omega)}(1)} \sum_{j=1}^{M} \sum_{s \in [2^j]^d} \langle f, \phi_{j,s} \rangle_{L_2(\Omega)} (b_{j,s} - \langle p, \phi_{j,s} \rangle_{L_2(\Omega)}) + \sum_{j=M+1}^{\infty} \sum_{s \in [2^j]^d} \langle f, \phi_{j,s} \rangle_{L_2(\Omega)} \langle p, \phi_{j,s} \rangle_{L_2(\Omega)}$$

$$= I_1 + I_2. \tag{A.6}$$

The term $I_2$ is the truncation error, which can be bounded by

$$I_2 \leq \left( \sum_{j=M+1}^{\infty} \sum_{s \in [2^j]^d} \langle f, \phi_{j,s} \rangle_{L_2(\Omega)}^2 \right)^{1/2} \left( \sum_{j=M+1}^{\infty} \sum_{s \in [2^j]^d} \langle p, \phi_{j,s} \rangle_{L_2(\Omega)} \right)^{1/2}$$

$$\leq 2^{-(Mdm_1 + Mdm_2)/2} \left( \sum_{j=M+1}^{\infty} 2^{jdm_2} \sum_{s \in [2^j]^d} \langle f, \phi_{j,s} \rangle_{L_2(\Omega)}^2 \right)^{1/2} \left( \sum_{j=M+1}^{\infty} 2^{jdm_1} \sum_{s \in [2^j]^d} \langle p, \phi_{j,s} \rangle_{L_2(\Omega)} \right)^{1/2}$$

$$\leq C_1 2^{-(Mdm_1 + Mdm_2)/2}, \tag{A.7}$$

where the first inequality is by the Cauchy-Schwarz inequality, and the last inequality is because $f \in \mathcal{W}^{m_2}(R_2)$ and $p \in \mathcal{W}^{m_1}(R_1)$.

Next, we consider $I_1$. It can be checked that

$$I_1 = \int_\Omega \sum_{j=1}^{M} \sum_{s \in [2^j]^d} \langle f, \phi_{j,s} \rangle_{L_2(\Omega)} \phi_{j,s} \mathrm{d}(\mu^* - \hat{\mu}_n). \tag{A.8}$$

Since $m_2 > d/2$, we can see that the function $\|f_M\|_{L_\infty} < R$ for all $M > 1$, where

$$f_M := \sum_{j=1}^{M} \sum_{s \in [2^j]^d} \langle f, \phi_{j,s} \rangle_{L_2(\Omega)} \phi_{j,s},$$

which, together with Lemma 5.11 of [86], implies that

$$\left| \int_\Omega f_M \mathrm{d}(\mu^* - \hat{\mu}_n) \right| = O_\mathbb{P}(n^{-1/2}), \tag{A.9}$$

where the right-hand-side term does not depend on $M$. Combining Eq.A.9 and Eq.A.8, the term $I_1$ can be bounded by

$$I_1 = O_\mathbb{P}(n^{-1/2}). \tag{A.10}$$

Plugging Eq.A.7 and Eq.A.10 into Eq.A.6, and noting that the right-hand-side term of Eq.A.10 does not depend on $M$, we can take $M \to \infty$ in Eq.A.7 to obtain

$$d_{\mathcal{F}_d}(\mu^*, \tilde{\mu}_n) = O_\mathbb{P}(n^{-1/2}), \tag{A.11}$$

which, together with Eq.A.3, leads to

$$-\lambda \int_\Omega \log \frac{p_n}{p^*} \mathrm{d}\hat{\mu}_n \le O_\mathbb{P}(n^{-1/2}). \tag{A.12}$$

By the triangle inequality and Eq.A.12, we obtain

$$-\lambda \int_\Omega \log \frac{p_n}{p^*} \mathrm{d}\mu^* \le O_\mathbb{P}(n^{-1/2}) + \lambda \left| \int_\Omega \log \frac{p_n}{p^*} \mathrm{d}(\mu^* - \hat{\mu}_n) \right|. \tag{A.13}$$

It remains to consider

$$\left| \int_\Omega \log \frac{p_n}{p^*} \mathrm{d}(\mu^* - \hat{\mu}_n) \right|,$$

which can be bounded by Lemma 5.11 of [86] again.

Specifically, since $\mathcal{G} \in \mathcal{W}^{m_2}(R_3)$, the entropy number of $\mathcal{G}$ can be bounded by [87]

$$H(u, \mathcal{G}, \|\cdot\|_P) \le C\delta^{-d/m_2},$$

where $C$ is a constant. Therefore, since $\mathcal{G} \subset \mathcal{W}^{m_2}(R_3)$, Lemma 5.11 of [86] gives us

$$\sup_{p \in \mathcal{G}} \left| \int_\Omega \log \frac{p}{p^*} \mathrm{d}(\mu^* - \hat{\mu}_n) \right| = O_\mathbb{P}(n^{-1/2}), \tag{A.14}$$

which, together with Eq.A.13, gives us

$$\mathrm{KL}(\mu^* \| \mu_n) = O_\mathbb{P}(\max\{n^{-1/2}, \lambda^{-1}n^{-1/2}\}).$$

By Eq.A.2, Eq.A.11, and Eq.A.14, we have

$$d_{\mathcal{F}_d}(\mu_n, \mu^*) \le 2d_{\mathcal{F}_d}(\mu^*, \tilde{\mu}_n) + \lambda \left| \int_\Omega \log \frac{p_n}{p^*} \mathrm{d}(P - P_n) \right| = O_\mathbb{P}(\max\{n^{-1/2}, \lambda n^{-1/2}\}). \tag{A.15}$$

Taking $\lambda$ as a constant finishes the proof.

# B  Estimating density of behavior policy using MaxEnt IRL is equivalent to training a GAN with specific likelihood function

In this section, we prove Proposition 3.1. In offline reinforcement learning scenario, the distribution of a trajectory $\tau = (s_0, a_0, s_1, a_1, ..., s_H, a_H) \in D$ can be written in the following form:

$$P_\theta(\tau) = P_{0_\mathcal{D}}(s_0) \prod_{t=0}^{H} \pi_\beta(a_t|s_t) T_\mathcal{D}(s_{t+1}|s_t, a_t). \tag{B.1}$$

In the training process, $P_{0_\mathcal{D}}(s_0), T_\mathcal{D}(s_{t+1}|s_t, a_t), \forall t \in [0, H]$ are known. Therefore, the uncertainty of the distribution of a trajectory is only related to the probability density of the behavior policy, and we have

$$P_\theta(\tau) = C \prod_{t=0}^{H} \pi_\beta(a_t|s_t) = CL_\theta^{\pi_\beta}(\tau), \tag{B.2}$$

where $L_\theta(\pi_\beta)$ is the likelihood function of $\pi_\beta$ given a trajectory, and $C$ is a constant with respect to dataset $\mathcal{D}$.

Following [71], in MaxEnt IRL we try to estimate the density of the trajectory by the Boltzmann distribution as

$$p_\theta(\tau) = \frac{1}{Z} \exp(-c_\theta(\tau)). \tag{B.3}$$

In [71], it has been shown that if we estimate $Z$ in the MaxEnt IRL formulation using guided cost learning, and suppose we have a GAN that can give an explicit density of the data, then optimizing the cost function of guided cost learning $\mathcal{L}_{cost}(\theta)$ is equivalent to optimizing the discriminator loss $\mathcal{L}_{discriminator}(D_\theta)$ in GAN. In the process of estimating $Z$, we also need to train a new sampling distribution $q(\tau)$ and use importance sampling to estimate $Z$. The new sampling policy is optimized by minimizing the KL divergence of $q(\tau)$ and the Boltzmann distribution in Eq.B.3:

$$\mathcal{L}_{sampler}(q) = \mathbb{E}_{\tau \sim q}[c_\theta(\tau)] + \mathbb{E}_{\tau \sim q}[\log(q(\tau))]. \tag{B.4}$$

The optimal sampling distribution is the demonstration policy (we call behavior policy in offline RL), which is an estimator of the behavior policy. In [71] setting, if we assume the sampling policy $q(\tau)$ is just an explicit density of GAN, we will show the generator loss that has a hybrid style as in Eq.4 is equivalent to optimize $\mathcal{L}_{sampler}(q)$. Note that

$$
\begin{aligned}
\mathcal{L}_{generator}(q) &= \mathbb{E}_{\tau \sim q}[(\log(1 - D_\tau)) - \log(D_\tau)] - \lambda \mathbb{E}_{\tau \sim q}[\log(q(\tau))] \\
&= \mathbb{E}_{\tau \sim q}[(\log \frac{q(\tau)}{\tilde{\mu}(\tau)} - \log \frac{\frac{1}{Z} \exp(-c_\theta(\tau))}{\tilde{\mu}(\tau)}] - \lambda \mathbb{E}_{\tau \sim q}[\log(q(\tau))] \\
&= \mathbb{E}_{\tau \sim q}[(\log(q(\tau)) - \log(\frac{1}{Z} \exp(-c_\theta(\tau))))] - \lambda \mathbb{E}_{\tau \sim q}[\log(q(\tau))] \\
&= \log(Z) + \mathbb{E}_{\tau \sim q}[c_\theta(\tau)] + \mathbb{E}_{\tau \sim q}[(1 - \lambda) \log(q(\tau))].
\end{aligned} \tag{B.5}
$$

In Eq.B.5, the term $\log(Z)$ is can be seen as a constant since it is just a normalizing term, so optimizing the generator loss $\mathcal{L}_{generator}(q)$ is equivalent to minimizing the KL divergence between $\tilde{C}q(\tau)$ and the Boltzmann distribution of the real trajectory density $p_\theta(\tau)$. Therefore, if we use a hybrid loss in the generator, the optimization of the generator is still equivalent to the optimization of $\mathcal{L}_{sampler}(q)$ up to a constant $\tilde{C}$. As $q(\tau) \propto \exp(-c_\theta(\tau)) \propto p_\theta(\tau)$, and combined with Eq.B.2, we have

$$L_\theta^{\pi_\beta}(\tau) = \frac{1}{C} P_\theta(\tau) \propto \exp(-c_\theta(\tau)) \propto q(\tau). \tag{B.6}$$

If we further denote the sampling policy estimated by GAN as $q_\theta^G(\tau)$, then we can get the result in Proposition 3.1.

## C  Details of the performance of the learning policy on the estimated product space of CPED

In this section, we will give a brief proof of Theorem 4.2, and show that the learning policy can find the optimal (state, action) w.r.t the training dataset $\mathcal{D}$.

Suppose the offline replay buffer is $\mathcal{D}$, with state space $\mathscr{S} = \{\mathcal{S}, \mathbb{P}_\mathcal{D}\}$ and action space $\mathscr{A} = \{\mathcal{A}, \pi_\beta\}$. Suppose the stationary point of the Bellman equation w.r.t the production sample space $\mathscr{S} \times \mathscr{A}$ is $Q^*(s^*, a^*)$. We consider two cases.

(1) If $\pi_\beta$ is an expert policy, then we have $(s^*, a^*) \in \mathcal{D}$, $Q(s^*, \pi_\beta(s^*)) = Q^*$. Thus, the optimal (state, action) pair appears on the seen area of dataset $\mathcal{D}$.

(2) If $\pi_\beta$ is an not an expert policy, then we have $(s^*, a^*) \notin \mathcal{D}$, $Q(s^*, \pi_\beta(s^*)) < Q^*$. In this situation, the optimal (state, action) pair appears on the unseen area of dataset $\mathcal{D}$, which belongs to the product space $\mathscr{S} \times \mathscr{A}$.

CPED mainly focuses on the second scenario, and we give proof of the performance of the learning policy when the optimal stationary point of the Bellman equation is in the unseen area of $\mathcal{D}$.

Let $\hat{\pi}(s) := \text{argmax}_a Q^*(s, a)$. We first show that $Q^*(s, a) = Q(s, \hat{\pi}^\Delta(s))$, where $\hat{\pi}^\Delta(s)$ is the probability-controlled learning policy using CPED that is defined on the support of behavior policy $\pi_\beta$. Suppose $\hat{\pi}^\Delta(s) > 0$ on $\mathscr{S} \times \mathscr{A}$ and $|Q(s, a)| \leq M, \forall (s, a) \in \mathscr{S} \times \mathscr{A}$. Then we have

$$
\begin{aligned}
Q^*(s, a) =& r(s, a) + \gamma \mathbb{E}_{s \sim \mathbb{P}_\mathcal{D}, a' \sim \pi^*}[Q^*(s', \pi^*(s'))] \\
\leq& r(s, a) + \gamma \mathbb{E}_{s \sim \mathbb{P}_\mathcal{D}, a' \sim \hat{\pi}}[Q^*(s', a')] \\
=& r(s, a) + \gamma \mathbb{E}_{s \sim \mathbb{P}_\mathcal{D}, a' \sim \hat{\pi}}[Q^*(s', a') 1_{(a' \in \Delta \pi_\beta(s'))}] + \gamma \mathbb{E}_{s \sim \mathbb{P}_\mathcal{D}, a' \sim \hat{\pi}}[Q^*(s', a') 1_{(a' \notin \Delta \pi_\beta(s'))}] \\
\leq& r(s, a) + \gamma \mathbb{E}_{s \sim \mathbb{P}_\mathcal{D}, a' \sim \hat{\pi}^\Delta}[Q^*(s', a')] + \gamma M \mathbb{E}_{s \sim \mathbb{P}_\mathcal{D}, a' \sim \hat{\pi}}[1_{(a' \notin \Delta \pi_\beta(s'))}] \\
=& \underbrace{r(s, a) + \gamma \mathbb{E}_{s \sim \mathbb{P}_\mathcal{D}, a' \sim \hat{\pi}^\Delta}[Q^*(s', a')]}_{I_1} + \underbrace{\gamma M \mathbb{P}((a' \notin \Delta \pi_\beta(s')))}_{I_2}
\end{aligned}
$$
(C.1)

Let action set $\Omega = \{a' : \pi_\beta(a'|s') = 0, \hat{\pi}(a'|s') > \epsilon, \forall s' \in \mathscr{S}\}$. If the conditional probability space $\pi_\beta$ can be estimated accurate, then by CPED, $\mathbb{P}(\Omega)$ is tending to zero. By Theorem 4.1, when using GAN with hybrid loss to estimate the density of $\pi_\beta$, with probability tending to one, we have $\mathbb{P}(\Omega) \to 0$. Then for term $I_2$,

$$
I_2 = \gamma M \mathbb{P}((a' \notin \Delta \pi_\beta(s'))) \leq \gamma M \mathbb{P}(\Omega) \to 0.
$$
(C.2)

For term $I_1$, based on iteration, we obtain

$$
\begin{aligned}
I_1 =& r(s, a) + \gamma \mathbb{E}_{s \sim \mathbb{P}_\mathcal{D}, a' \sim \hat{\pi}^\Delta}[r(s', a') + \gamma \mathbb{E}_{s \sim \mathbb{P}_\mathcal{D}, a'' \sim \pi^*}[Q^*(s'', \pi^*(s''))]] \\
\leq& r(s, a) + \gamma \mathbb{E}_{s \sim \mathbb{P}_\mathcal{D}, a' \sim \hat{\pi}^\Delta}[I_1' + \gamma M \mathbb{P}(\Omega)] \\
\leq& r(s, a) + \gamma \mathbb{E}_{s \sim \mathbb{P}_\mathcal{D}, a' \sim \hat{\pi}^\Delta}[r(s', a') + \gamma \mathbb{E}_{s \sim \mathbb{P}_\mathcal{D}, a' \sim \hat{\pi}^\Delta}[I_1'' + \gamma M \mathbb{P}(\Omega)] + \gamma M \mathbb{P}(\Omega)] \\
\leq& \mathbb{E}[r(s, a) + \gamma r(s', \hat{\pi}^\Delta(s')) + \gamma^2 r(s'', \hat{\pi}^\Delta(s'')) + \ldots] + (\gamma + \gamma^2 + \gamma^3 + \ldots) M \mathbb{P}(\Omega) \\
=& Q(s, \hat{\pi}^\Delta(s)) + \frac{\gamma}{1 - \gamma} M \mathbb{P}(\Omega).
\end{aligned}
$$
(C.3)

Since $\mathbb{P}(\Omega) \to 0$, combining Eq.C.1 and Eq.C.3 yields

$$
\begin{aligned}
Q^*(s, a) \leq& I_1 + I_2 \\
\leq& Q(s, \hat{\pi}^\Delta(s)) + \frac{\gamma}{1 - \gamma} M \mathbb{P}(\Omega) + \gamma M \mathbb{P}(\Omega) \to Q(s, \hat{\pi}^\Delta(s))
\end{aligned}
$$
(C.4)

On the other hand, obviously $Q^*(s, a) \geq Q(s, \hat{\pi}^\Delta(s))$, a.s..

Hence, with probability tending to one, $Q^*(s, a)$ is close to $Q(s, \hat{\pi}^\Delta(s))$.

Then the learning policy $\hat{\pi}^\Delta$ trained by CPED is close to the optimal policy in product space $\mathscr{S} \times \mathscr{A}$ with probability tending to one.

# D   Details of the convergence learning policy of CPED when using policy iteration

In this section, we will give a brief proof of Theorem 4.3, and show the convergence of the learning policy when using policy iteration to update the learning policy. The whole proof is divided into two parts. First, we show the monotonic improvement of Q function of the iterated policy by CPED. Then we give the convergence of the iterated value function.

## D.1   Monotonic improvement of Q function $Q^{\pi_{k+1}^\Delta}(s, a) \geq Q^{\pi_k^\Delta}(s, a)$

*Proof.* Suppose we use iteration to improve our policy in each training step by

$$
\pi_{k+1} := \underset{a}{\text{argmax}} \, Q^{\pi_k}(s, a), \forall s \in \mathscr{S}
$$
(D.1)

Then at iteration $k+1$, we have

$$
\begin{aligned}
Q^{\pi_{k+1}}(s,a) - Q^{\pi_k}(s,a) =& \gamma \mathbb{E}_{s'\sim\mathbb{P}_\mathcal{D}}[Q^{\pi_{k+1}}(s',\max_{a'_{k+1}}\pi_{k+1}(s')) - Q^{\pi_k}(s',\max_{a'_k}\pi_k(s'))] \\
=& \gamma \mathbb{E}_{s'\sim\mathbb{P}_\mathcal{D}}[Q^{\pi_{k+1}}(s',\max_{a'_{k+1}}\pi_{k+1}(s')) - Q^{\pi_k}(s',\max_{a'_{k+1}}\pi_{k+1}(s'))+ \\
& \underbrace{Q^{\pi_k}(s',\max_{a'_{k+1}}\pi_{k+1}(s')) - Q^{\pi_k}(s',\max_{a'_k}\pi_k(s'))]}_{\geq 0,\text{by definition of policy iteration}} \\
\geq& \gamma \mathbb{E}_{s'\sim\mathbb{P}_\mathcal{D}}[Q^{\pi_{k+1}}(s',\max_{a'_{k+1}}\pi_{k+1}(s')) - Q^{\pi_k}(s',\max_{a'_{k+1}}\pi_{k+1}(s'))] \\
\geq& \gamma \mathbb{E}_{s'\sim\mathbb{P}_\mathcal{D},a'_{k+1}\sim\pi_{k+1}}[Q^{\pi_{k+1}}(s',a'_{k+1}) - Q^{\pi_k}(s',a'_{k+1})] \\
=& \underbrace{\gamma \mathbb{E}_{s'\sim\mathbb{P}_\mathcal{D},a'_{k+1}\sim\pi_{k+1}}[(Q^{\pi_{k+1}}(s',a'_{k+1}) - Q^{\pi_k}(s',a'_{k+1}))1_{(a'_{k+1}\in\Delta\pi_\beta(s))}]}_{I_1} + \\
& \underbrace{\gamma \mathbb{E}_{s'\sim\mathbb{P}_\mathcal{D},a'_{k+1}\sim\pi_{k+1}}[(Q^{\pi_{k+1}}(s',a'_{k+1}) - Q^{\pi_k}(s',a'_{k+1}))1_{(a'_{k+1}\notin\Delta\pi_\beta(s))}]}_{I_2}.
\end{aligned}
\tag{D.2}
$$

For term $I_2$, let $X_{k+1} = (Q^{\pi_{k+1}}(s',a'_{k+1}) - Q^{\pi_k}(s',a'_{k+1}))1_{(a'_{k+1}\notin\Delta\pi_\beta(s))}$. Then it holds that

(1) $X_{k+1} \to 0$ in probability, as $1_{(a'_{k+1}\notin\Delta\pi_\beta(s))} \to 0$ by policy control defined in CPED.

(2) $X_{k+1}$ is bounded by some constant.

So based on the Dominated Convergence Theorem, $\mathbb{E}X_{k+1} \to 0$.

For term $I_1$, we have

$$
\begin{aligned}
I_1 =& \gamma \mathbb{E}_{s'\sim\mathbb{P}_\mathcal{D},a'_{k+1}\sim\pi_{k+1}}[(Q^{\pi_{k+1}}(s',a'_{k+1}) - Q^{\pi_k}(s',a'_{k+1}))1_{(a'_{k+1}\in\Delta\pi_\beta(s))}] \tag{D.3} \\
=& \gamma \mathbb{E}_{s'\sim\mathbb{P}_\mathcal{D},a'_{k+1}\sim\pi^\Delta_{k+1}}[Q^{\pi_{k+1}}(s',a'_{k+1}) - Q^{\pi_k}(s',a'_{k+1})]. \tag{D.4}
\end{aligned}
$$

By iteration,

$$
\begin{aligned}
& (Q^{\pi_{k+1}}(s',a'_{k+1}) - Q^{\pi_k}(s',a'_{k+1}))1_{(a'_{k+1}\in\Delta\pi_\beta(s))} \\
& \geq \gamma \mathbb{E}_{s''\sim\mathbb{P}_\mathcal{D},a''_{k+1}\sim\pi^\Delta_{k+1}}[Q^{\pi_{k+1}}(s'',a''_{k+1}) - Q^{\pi_k}(s'',a''_{k+1})] + \gamma I_2. \tag{D.5}
\end{aligned}
$$

So we have

$$
Q^{\pi_{k+1}}(s,a) - Q^{\pi_k}(s,a) \geq \gamma^\infty + \frac{\gamma}{1-\gamma}I_2 \to 0 \quad when \quad a \in \Delta\pi_\beta. \tag{D.6}
$$

Therefore, for sufficiently large $k$, $Q^{\pi^\Delta_{k+1}}(s,a) \geq Q^{\pi^\Delta_k}(s,a)$. Furthermore, we have $V^{\pi^\Delta_{k+1}}(s) \geq V^{\pi^\Delta_k}(s)$. $\qquad\square$

## D.2 Convergence of the iterated value function $\|V^{\hat\pi^\Delta_{k+1}} - V^*\|_\infty \leq \gamma^{k+1}\|V^{\hat\pi^\Delta_0} - V^*\|_\infty$

*Proof.* Direct computation shows that

$$
\begin{aligned}
V^*(s) - V^{\hat\pi^\Delta_{k+1}}(s) =& \max_a[r(s,a) + \gamma\mathbb{E}_{s'\sim\mathbb{P}_\mathcal{D}}V^*(s')] - [r(s,\hat\pi^\Delta_{k+1}(s)) + \gamma\mathbb{E}_{s'\sim\mathbb{P}_\mathcal{D}}V^{\hat\pi^\Delta_{k+1}}(s')] \\
\leq& \max_a[r(s,a) + \gamma\mathbb{E}_{s'\sim\mathbb{P}_\mathcal{D}}V^*(s')] - [r(s,\hat\pi^\Delta_{k+1}(s)) + \gamma\mathbb{E}_{s'\sim\mathbb{P}_\mathcal{D}}V^{\hat\pi^\Delta_k}(s') \\
=& \max_a[r(s,a) + \gamma\mathbb{E}_{s'\sim\mathbb{P}_\mathcal{D}}V^*(s')] - \max_a[r(s,a) + \gamma\mathbb{E}_{s'\sim\mathbb{P}_\mathcal{D}}V^{\hat\pi^\Delta_k}(s')] \\
\leq& \max_a(r(s,a) + \gamma\mathbb{E}_{s'\sim\mathbb{P}_\mathcal{D}}V^*(s') - (r(s,a) + \gamma\mathbb{E}_{s'\sim\mathbb{P}_\mathcal{D}}V^{\hat\pi^\Delta_k}(s'))) \\
\leq& \gamma\|V^* - V^{\hat\pi^\Delta_k}\|_\infty \leq \gamma^2\|V^* - V^{\hat\pi^\Delta_{k-1}}\|_\infty \leq \cdots \leq \gamma^{k+1}\|V^{\hat\pi^\Delta_0} - V^*\|_\infty.
\end{aligned}
\tag{D.7}
$$

Since Eq.D.7 holds for $\forall s \in \mathscr{S}$, we have

$$\|V^{\hat{\pi}_{k+1}^\Delta} - V^*\|_\infty \le \gamma^{k+1}\|V^{\hat{\pi}_0^\Delta} - V^*\|_\infty.$$

This finishes the proof. □

# E  Experiment Details and More Results

## E.1  The Mujoco and AntMaze tasks

- **Gym-MuJoCo Task.** The Gym-MuJoCo is a commonly used benchmark for offline RL task. Each Gym-MuJoCo task contains three types of static offline datasets, generated by different behavior policies. The three offline datasets include: **i)** Random dataset originates from random policies and contains the least valuable information **ii)** The medium-quality dataset is generated by a partially trained policy (which performs about 1/3 as well as an expert policy).The medium-quality dataset contains three types of mixture: medium, medium-replay,medium-expert. **iii)** The expert dataset is produced by a policy trained to completion with SAC.

  In offline RL, the datasets generated by the medium-quality policy are better considered since we are confronted with such dataset in most practical scenarios. Therefore, in our experiments, we consider the datasets from the medium-quality policy.

- **AntMaze Task.** The AntMaze task is a challenging navigation task that requires a combination of different sub-optimal trajectories to find the optimal path. According to different task levels, the AntMaze task is divided into umaze, medium, and large types.

## E.2  Implementation Details and Discussion

Implementation of CPED algorithm contains Flow-GAN training and Actor-Critic training.

**Flow-GAN training.** The generator of GAN follows the NICE [34] architecture of the flow model, and the discriminator is a 4-layer MLPs. The entire code of training Flow-GAN is based on the implementation of FlowGAN[33][4]. FlowGAN[33] is intended to deal with image data, while the D4RL dataset does not contain image features. So in our training, we replace the image data-specific network architecture(such as convolution layers, Residual block) with fully connected layers. The hyperparameters used in Flow-GAN is shown in Table E.1.

**Actor-Critic training.** The Actor-Critic training implementation follows TD3[45], as SPOT[22] suggested TD3 architecture performs better than SAC in offline RL tasks. Additionally, the reward in AntMaze tasks is centered following the implementation of CQL/IQL. When training CPED in practice, the Flow-GAN updates first after initializing the CPED. The policy and $Q$ networks start to update when Flow-GAN has trained 20 thousand steps, and a relatively reliable density estimator is provided. Afterward, the Actor-Critic begins to train simultaneously. To prevent overfitting, the Flow-GAN stops training when it becomes stable.The hyperparameters used in Actor-Critic training are shown in Table E.2.

**Hyperparameter tuning of the time(epoch) varying $\alpha$.** The key element in the time-varying setting of policy constraint parameter $\alpha$ is to find the task's strongest and weakest $\alpha$ values. We follow the pattern we observed in automatic learning of $\alpha$ in BEAR[17] and set the strongest and weakest $\alpha$ values for different offline RL tasks. We find that after 200-300 thousand steps of learning with the strongest policy constraint, we get a relatively strong learning policy. Then we decrease $\alpha$ to the weakest value. Indeed, this decreasing process is robust to small changes of $\alpha$. The settings in Figure 2(b) and 2(c) are generally applicable for most tasks in Mujoco task and AntMaze task.

## E.3  Training Curve

The training curve of Mujoco tasks is shown in Figure E.1. The training curve of AntMaze tasks is shown in Figure E.2.

---

[4]https://github.com/ermongroup/flow-gan

Table E.1: The hyperparameter used in Flow-GAN training.

| | Hyperparameter | Value |
|---|---|---|
| Shared parameter | Optimizer | Adam |
| | Batch size | 256 |
| | Gradient penalty for WGAN-GP[59] | 0.5 |
| | Training ratio of the generator and discriminator | 5:1 for DCGAN[60] 1:5 for WGAN-GP |
| | Maximium Likelihood Estimation(MLE) adjustment weight | 1 |
| | Activation in the hidden layer | LeakyRelu |
| Generator | Learning rate | 1e-4 |
| | Latent layer of m network in NICE[34] | 3 |
| | Latent dim | 750 |
| | Hidden Layer | 4 |
| | Output activation | LeakyRelu |
| Discriminator | Learning rate | 1e-5 for DCGAN 1e-4 for WGAN-GP |
| | Latent dim | [2 × (action dim +state dim), 4 × (action dim +state dim)] |
| | Hidden Layer | 2 |
| | Normalization | Batch Norm for DCGAN Layer Norm for WGAN-GP |
| | Output activation | Sigmoid for DCGAN Identity for WGAN-GP |
| | Dropout rate | 0.2 for DCGAN 0.0 for WGAN-GP |

Table E.2: The hyperparameters used in Actor-Critic training.

| | Hyperparameter | Value |
|---|---|---|
| TD3 | Optimizer | Adam |
| | Batch size | 256 |
| | Actor learning rate | 3e-4 |
| | Critic learning rate | 3e-4 |
| | Discount factor | 0.99 |
| | Number of iterations | 1e6 |
| | Target update rate $\tau$ | 0.005 |
| | Policy noise | 0.2 |
| | Policy noise clipping | 0.5 |
| | Policy update frequency | 2 |
| Architecture | Actor hidden dim | 256 |
| | Actor layers | 3 |
| | Critic hidden dim | 256 |
| | Critic layers | 3 |
| CPED | Time varying $\alpha$ | see in Figure 2(b)-(c) |

## E.4 Target Q function during training

We show the target Q function of different Mujoco Tasks in Figure E.3 and give an experimental analysis of the policy constraint method. The target Q function of the AntMaze task is quite small due to the sparse reward, and we only show the target Q function of the Mujoco Task. In the offline RL scenario, the target Q function in the Bellman equation is vulnerable to OOD points. For OOD points, the target Q function are overestimated, and the Q value could be very large when the extrapolation errors are accumulated continuously. As a consequence, the learning policy fails to predict a reasonable action for most cases. CQL[13] discusses that policy control alone can not prevent the target Q function from being overestimated on OOD points. The experiment shows if we

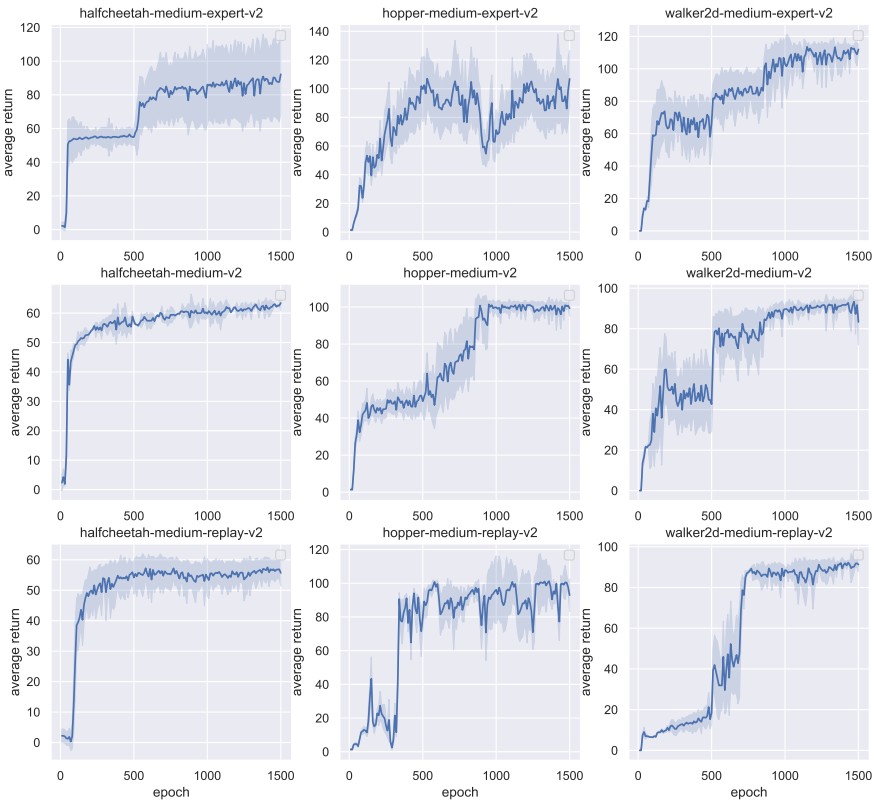

Figure E.1: Training curve of different Mujoco Tasks. All results are averaged across 5 random seeds. Each epoch contains 1000 training steps.

can get an explicit and relatively accurate density function of the behavior policy, we can get a stable and meaningful target Q function.

## E.5   Ablations

In the ablation study, we compare the performances of two different settings of $\epsilon$ in Eq.7, in which $\epsilon$ is set to 0 (traditional quantile setting ) and batch mean likelihood (our setting), respectively[17, 22]. The learning curves of these two settings are shown in Figure E.4. We only show the performance of the Mujoco task as AntMaze tasks are very unstable in the ablation study. For most Mujoco tasks, the batch mean likelihood setting for $\epsilon$ achieves higher normalized return, while the learning cure of the traditional quantile setting is quite volatile.

We further analyze the time-varying constraint setting we used in CPED by comparing its performance with that of the constant constraint setting. To better show the differences between two kinds of constraint settings on $\alpha$, we take the maximum and minimum values of the time-varying constraint as the constant constraint value. The learning curves of these two settings are shown in Figure E.5. We only show the Mujoco task performance as AntMaze tasks are unstable. For most Mujoco tasks, the time-varying constraint setting of $\alpha$ outperforms the constant setting with a large margin.

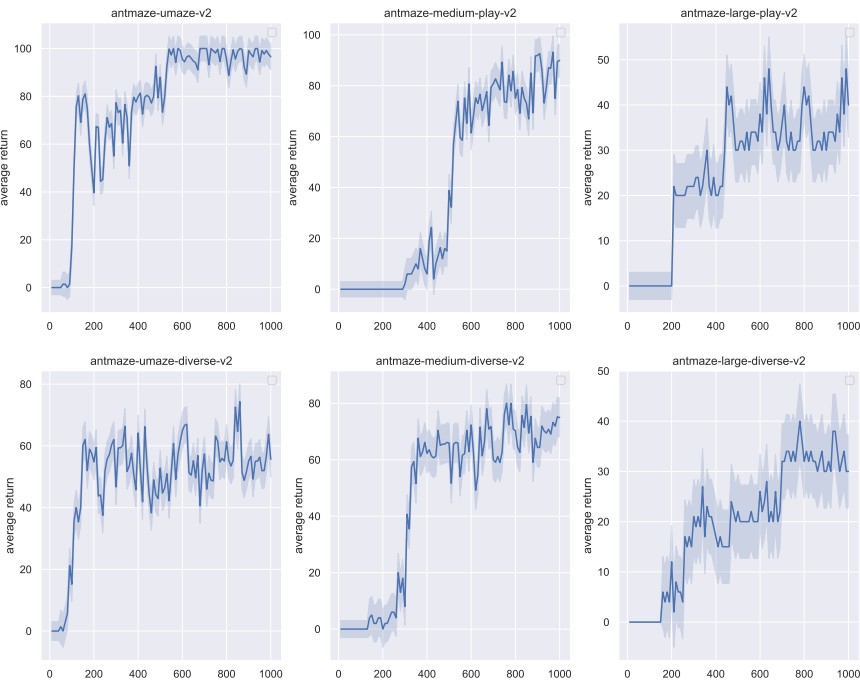

Figure E.2: Training curve of different Antmaze Tasks. All results are averaged across 5 random seeds. Each epoch contains 1000 training steps.

We also used the time-varying constraint trick for Spot and randomly selected three tasks to compare the performance difference between Spot using its own constant alpha and the time-varying alpha setting (Figure E.6). We can see that the performances with time-varying $\alpha$ are close or even inferior to those with constant $\alpha$ setting, indicating the trick of varying does not result in too much benefit.

## E.6  Density estimation accuracy by Flow-GAN and VAE: a toy example

To assess the accuracy of Flow-GAN for distribution estimation, this section conducts a straightforward toy experiment. The experiment involves a comparison between the performance of Flow-GAN and VAE in learning Gaussian mixture distributions and generating samples. Specifically, two Gaussian distribution settings were employed in the experiment:

- We gather approximately $12800000$ data points from a multivariate Gaussian distribution with a mean of $\mu = [1, 9]$ and covariance matrix of $\Sigma = [[1, 0], [0, 1]]$.
- We gather a similar amount (with setting 1) data points from a Gaussian mixture distribution, in which data are randomly drawn from two independent Gaussian distributions ($\mu_1 = [1, 1]$ and $\mu_2 = [9, 9]$, $\Sigma_1 = \Sigma_2 = [[1, 0], [0, 1]]$) with equal probability.

We then employ both VAE and Flow-GAN to generate samples and calculate the mean of log-likelihood. Table E.3 demonstrates that Flow-GAN outperforms VAE significantly in approximating the original data distribution. So, as many previous studies [29, 30, 31, 32] have shown, VAE has shortcomings in estimating complex distributions, especially multimodal distributions or complex behavior policy distributions.Flow-GAN has a stronger advantage in learning complex distributions due to the use of MLE and GAN-based adversary loss.

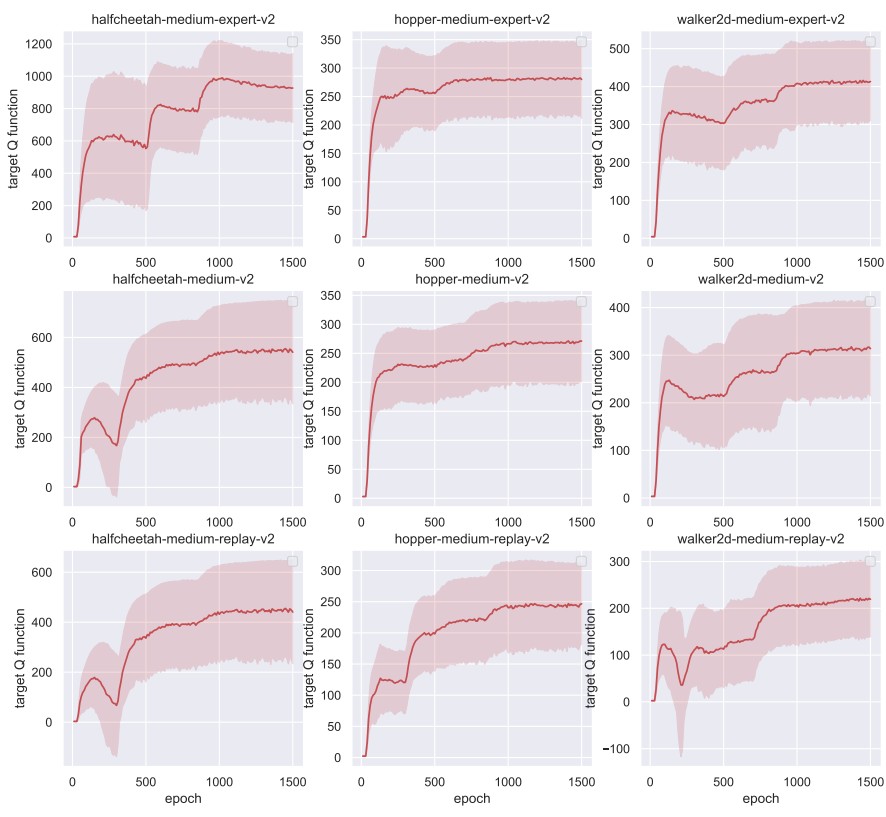

Figure E.3: Target Q function of different Mujoco Tasks. All results are averaged across 5 random seeds. Each epoch contains 1000 training steps.

Table E.3: The mean of log-likelihood of the generated sample by Flow-GAN and VAE.

| Setting | Ground Truth | VAE | Flow-GAN |
|---------|--------------|---------|----------|
| setting1 | -2.89 | -140.83 | -24.18 |
| setting2 | -2.78 | -50.05 | -6.12 |

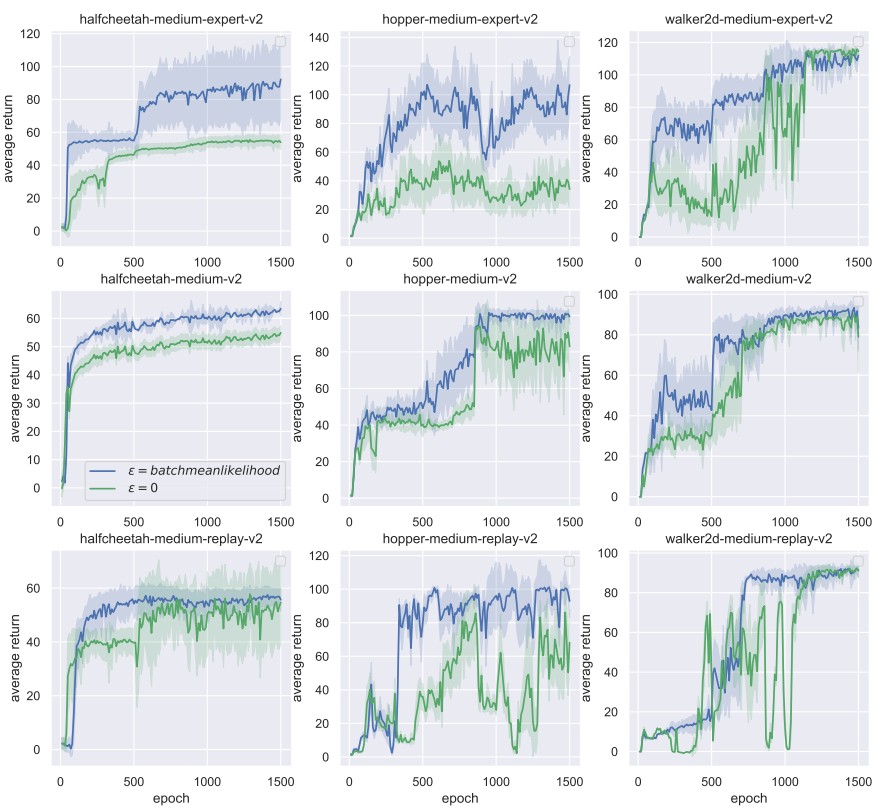

Figure E.4: Training curve of different Mujoco Tasks when using different likelihood threshold $\epsilon$. All results are averaged across 5 random seeds. Each epoch contains 1000 training steps. The green line denotes the training curve under $\epsilon = 0$, and the blue line denotes the training curve under our batch mean likelihood setting.

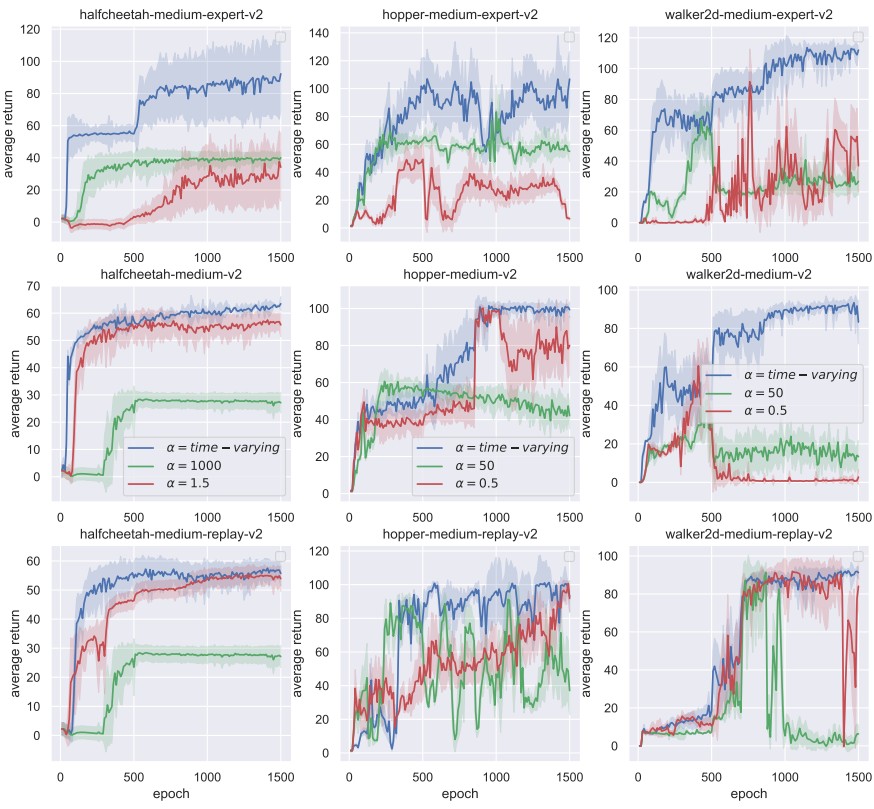

Figure E.5: Training curve of different Mujoco Tasks when using different constraint threshold $\alpha$. All results are averaged across 5 random seeds. Each epoch contains 1000 training steps. For each task (halfcheetah/hopper/walker2d), the selected $\alpha$ in the experiment remains the same under three types of dataset (medium-expert-v2/medium-v2/medium-replay-v2).

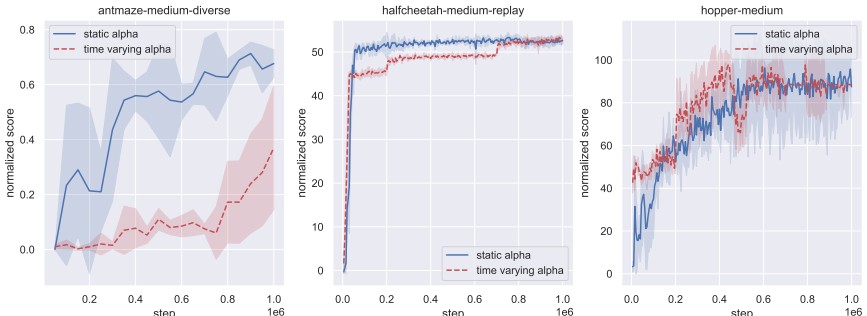

Figure E.6: Training curve of three randomly chosen tasks for spot when using constant $\alpha$ and the time-varying $\alpha$ setting. All results are averaged across 5 random seeds. Each epoch contains 1000 training steps. The blue line denotes the training curve using SPOT's constant $\alpha$ setting, and the red line denotes the training curve using time-varying $\alpha$ setting.