# OpenReview forum: "Constrained Policy Optimization with Explicit Behavior Density For Offline Reinforcement Learning"
_NeurIPS.cc/2023/Conference — NeurIPS 2023 poster_

### Official Review · Reviewer_krNs · 2023-07-02

**Soundness:** 3 good
**Presentation:** 3 good
**Contribution:** 3 good
**Rating:** 6
**Confidence:** 4

**Summary:**

The paper proposes a novel algorithm for offline model-free reinforcement learning by constraining the actor to lie within a safe area defined by an explicit behavioral density. This behavioral density is obtained by thresholding a FlowGAN model. The resulting algorithm is competitive across various D4RL datasets.


**Strengths:**

- Clearly presented algorithm and well-motivated method
- Strong empirical evaluation and competitive results on D4RL MuJoCo
- Solid theoretical justification of the convergence of the FlowGAN and RL training


**Weaknesses:**

- Model-based offline methods typically use a state-action-based uncertainty penalty which could also be repurposed for the definition of the safe area. E.g. thresholding the model uncertainty as in MOReL [1]. This would be a useful baseline to show the benefit of the Flow-GAN method
- The hand-crafted per-environment piecewise linear scheme for alpha limits the generality of the method
- Algorithm 1 seems to be incorrect, the preceding discussion has the Flow-GAN optimization happening before RL training, whereas Algorithm 1 has the Flow-GAN optimization interleaved with RL training.
- Line 150: definition of the ‘offline MDP’ is not a real MDP. For example, the state and action spaces should both be sets and the definition is a tuple. It is unclear what this definition adds beyond describing the marginal state distribution and behavioral policy.
- In Theorem 4.2, the optimal Q-function on the “restricted product space” needs to be defined and explained.

Minor:
- Line 145: ‘triples’ is likely a mistake, the transition is a 4-tuple
- Tables 1 and 2 missing EDAC [2], the current SOTA model-free offline RL method

[1] MOReL: Model-Based Offline Reinforcement Learning. Rahul Kidambi, Aravind Rajeswaran, Praneeth Netrapalli, Thorsten Joachims.

[2] Uncertainty-Based Offline Reinforcement Learning with Diversified Q-Ensemble. Gaon An, Seungyong Moon, Jang-Hyun Kim, Hyun Oh Song.


**Questions:**

- Could a dynamics model-based uncertainty (e.g. MOReL) provide a simple alternative to the Flow-GAN threshold?
- Could the alpha tuning scheme be replaced by something not hand-crafted to increase the applicability of the method?


**Limitations:**

Not discussed in the paper.

---

> ### Author Rebuttal · Authors · 2023-08-08
>
> Thank you for your review. We appreciate your kind suggestions for improving our manuscript.
>
> **Response to Weakness 1**:  We agree that MORel is a useful baseline for CPED, and we will certainly include the MORel[1] in our manuscript as baseline method. From our understanding, MoReL provides us good estimates of the offline RL dynamics, enabling us to have a better understanding of the safety of the state space as well. In this sense, both CPED and MORel want to identify safe areas, while they implement in different approaches.
>
> **Response to Weakness 2**:  Among policy control methods for solving offline RL problems, some works (e.g., BEAR, UWAC) determine the value of alpha through automated learning (Lagrangian duality optimization). However, BRAC (ref[1]) has demonstrated through experimental studies that the performance of automatically optimized alpha is not as effective as manually determined alpha values. Therefore, the CPED selects the hand-craft threshold alpha. To better control the training process, CPED further proposes the piecewise linear scheme for adjusting alpha (To our best knowledge, the CPED is the first one using a piecewise linear scheme for determining alpha in offline RL solutions). In section 5.2 and Appendix E.5, we present the details of setting the piecewise linear alpha, and the experiments show that this scheme delivers better returns in most scenarios.
>
>
> **Response to Weakness 3:**  We appreciate you pointing out the issue of training order when training Flow-GAN and RL models. When implementing the CPED, the Flow-GAN is joined trained with RL model. Specifically, the Flow-GAN is first trained with several epochs until it becomes stable, then the RL model is interleaved trained with Flow-GAN.
>
> Considering the Flow-GAN starts training first, in the manuscript, the description of Flow-GAN appears before the RL model. We will further revise Algorithm 1 and clearly state that the Flow-GAN model is firstly trained for a few epochs (M epochs), and then interleaved trained with RL model. Please find the attached **pdf file** ( in “**Author Rebuttal by Authors**” part at the top of this page) for the revised figure of Algorithm 1.
>
>
> **Response to Weakness 4**: Thank you for pointing out the unclear statement in the manuscript. The offline MDP is not real MDP as it is under online setting. The action/state under offline MDP is bounded in the probability space. To better illustrate the concept of “offline MDP”, we have revised the definition in Line 150. Please find the attached **pdf file** ( in “**Author Rebuttal by Authors**” part at the top of this page) for the revised Definition of “Offline MDP”
>
> We want to further explain that the main purpose of proposing the definition of offline MDP is to highlight the distinction between offline RL and online RL. In offline RL, the state and action spaces are bounded, and this boundary is constrained by the probability measure defined over these spaces. Therefore, in subsequent theoretical proofs and algorithm designs, we work within this bounded state and action space.
>
>
> **Response to Weakness 5**:  Actually, we have explained that “restricted product space” in theorem 4.2 refers to ${\mathscr{S} \times \mathscr{A}}$, where \mathscr{S} and \mathscr{A} are measured state space and measured action space defined in “Offline MDP” in Line 150 respectively.
>
> **Response to Minor Issue 1**: We really appreciate you for pointing out the grammar error. We will further revise the sentence as “Dataset $\mathcal{D}$ contains many 4-tuples $(s,a,s',r)$ that can be viewed as independent.”
>
> **Response to Minor Issue 2**: Thank you for pointing out the related method and references. We will certainty include the above references and the EDAC methods in the revised manuscript (we have cited [2] already). From Table 1 in [2], the performance of EDAC is competitive with the proposed CPED and both methods contribute to the SOTA methods for offline RL task.
>
> It is also noted that multiple Q-functions are utilized in EDAC method, while there are only 2 Q-networks in CPED. Thus, motivated by [2], we believe the proposed CPED has space for further improvement when multiple Q-functions are applied.
>
> **Response to Question 1**: We appreciate this interesting question. As far as we know, model-based uncertainty methods (e.g., MOReL) focus more on state space when defining safe regions, while CPED pays more attention to the safety of the action space. In fact, during the training phase of offline RL method, OOD actions are the main source of model failure, whereas OOD states are more critical during the testing phase. Though MoReL also considers the uncertainty of action space in its dynamics model-based uncertainty, our view on whether MoReL can provide an alternative to Flow-GAN in this regard is relatively conservative.
>
> Furthermore, we believe that the key to defining a safe area lies in the ability to estimate the distribution of the safe area well. MoReL uses Gaussian dynamics models, and whether it can be used as alternative for flow-GAN threshold depends on whether the Gaussian dynamics models have enough representational power to learn complex distributions.
>
> **Response to Question 2**:  As we mentioned in Question 2, we can apply the Lagrangian duality optimization for updating alpha to increase the applicability of CPED (e.g. BEAR and UWAC). Nevertheless, according to BRAC (ref[1]), this updating strategy may not lead to a good model performance. Thus, we select the hand-craft alpha in CPED when implementing the method. Especially we further propose piecewise linear scheme to determine alpha dynamically, and the effectiveness of the piecewise linear scheme is also validated from experiments.
>
> ref[1]： https://arxiv.org/abs/1911.11361

---

> > ### Comment · Reviewer_krNs · 2023-08-13
> > **Thanks for the response**
> >
> > Thank you for the responses.
> >
> > W2. I don't think a handcrafted schedule should be presented as an advantage. Any other algorithm could have manually done this but it would have reduced the applicability of their method.
> >
> > W3. Could the authors clarify why it is interleaved? Why would it not be better to use a GAN trained to convergence?
> >
> > Q1: The dynamics model in MOReL provides an uncertainty bonus over both state and action, so may already incorporate action uncertainty.

---

> > > ### Author Response · Authors · 2023-08-14
> > > **Thanks for your comments**
> > >
> > > **W2. I don't think a handcrafted schedule should be presented as an advantage. Any other algorithm could have manually done this but it would have reduced the applicability of their method.**
> > >
> > > **Response:** Thank you for your comment. We agree with you that handcrafted alpha may increase the difficulties in hyperparameter tuning, thus limiting the method's applicability. However, our piece-wise alpha scheme is straightforward to implement. Specifcally, we would like to further emphasize the following:
> > >
> > > 1. While the idea of automatic learning alpha (optimizing Lagrangian multiplier) appears in some early works (BEAR/UWAC), more recent papers(BRAC) tend to prefer hand-craft alpha based on experimental considerations.
> > >
> > > 2. We agree that the handcraft scheme of alpha is not the specific advantage of our method. (The advantage of our manuscript comes from the Flow-GAN model). As a matter of fact, the hand-craft scheme is a userful trick in model training for better convergence and results.
> > >
> > >
> > > 3. In our manuscript, the piece-wise linear (constant) scheme is suggested, and this scheme is easy to implement. We only need to set continuous intervals (4 intervals in Mujoco and Antmze tasks, actually not very sensitive from experiments, see Figure 2(a,b) in the manuscript), and the alpha for each interval decreases exponentially during the training process.
> > >
> > >
> > > **W3. Could the authors clarify why it is interleaved? Why would it not be better to use a GAN trained to convergence?**
> > >
> > > **Response:** Thank you for your comment. We are sorry for our unclear statement in our last response.
> > >
> > > First, we would like to clarify that before conducting the offline RL task, the GAN has undergone training to achieve convergence (when Flow-GAM becomes stable). Subsequently, the GAN is trained in an interleaved manner with the RL model. The reason for interleaving the training of the Flow-GAN and RL model is to enhance the Flow-GAN's adaptation to downstream offline RL tasks.
> > >
> > > Actually, we have attempted only training  the RL model with a converged and fixed Flow-GAN, we observed that the convergence of the RL model was not sufficiently stable. Therefore, we recommend trying the interleaved training approach.
> > >
> > >
> > > **Q1: The dynamics model in MOReL provides an uncertainty bonus over both state and action, so may already incorporate action uncertainty.**
> > >
> > > **Response :** Thank you for pointing out the merits of MORel. We acknowledge that MORel captures the uncertainties in both action and state, which we also mentioned in our previous response.(*Though MoReL also considers the uncertainty of action space....*).
> > >
> > > As for whether MORel can be a substitute for the flow-GAN threshold, its feasibility relies on its capacity to learn complex distributions. We find this to be an interesting topic (along with the utilization of model-based mechanisms to assist model-free methods), and it would be meaningful to delve deeper into this direction in the future. We again appreciate your helpful suggestions.

---

### Official Review · Reviewer_1U1v · 2023-07-04

**Soundness:** 3 good
**Presentation:** 3 good
**Contribution:** 2 fair
**Rating:** 5
**Confidence:** 4

**Summary:**

This paper proposes a novel approach named Constrained Policy optimization with Explicit Behavior density (CPED) to mitigate the existing policy control methods, e.g., overly conservative or failing to identify OOD areas accurately. CPED use a flow-GAN model to explicitly estimate behavior policy density. The strength of CPED lies in its ability to accurately identify safe regions for explorations, which in turn, leads to less conservative learning policies.

**Strengths:**

1. Well-motivated example in Fig. 1 and overall good writing.
2. Sufficient theoretical evidence to show that CPED has a fast convergence rate and is able to ﬁnd the optimal Q-function value.

**Weaknesses:**

1. The contribution is limited as it seems to directly borrow the idea of flow-gan into offline RL domains.
2. Besides the motivation example in Fig. 1, there is no faithful evidence to show that "accurately identify the feasible region, which includes both observed and unobserved but safe points" both in theoretical analysis and experiments.
3. Lack of survey and comparison with the most similar work [1], which also learns an expressive generative behavior model but by the diffusion technique.
4. Though there is an obvious edge of CPED against SPOT in experiments, more explanations are lacking to suppose it.


**Questions:**

1. Could the authors provide more comparisons between CPED and SPOT and explain why CPED is more feasible?
2. Regarding policy learning (Eq. 7), why not transfer it to a Lagrange function like SPOT? To the best of my knowledge, explicitly excluding bad transitions has a little edge against other weighting or penalty methods in terms of generalization, which is a key contribution mentioned by the authors.


**Limitations:**

Lack of related work: the in-sample learning paradigm [2-8] is now one of the trends in offline reinforcement learning, those new methods are worth being well mentioned in related work.

---
I decided to change my score from 4 to 5 after the first round discussion with the authors.

---

[1] Chen H, Lu C, Ying C, et al. Offline reinforcement learning via high-fidelity generative behavior modeling[J]. arXiv preprint arXiv:2209.14548, 2022.

[2] Kostrikov I, Nair A, Levine S. Offline reinforcement learning with implicit q-learning[J]. arXiv preprint arXiv:2110.06169, 2021.

[3] Xu H, Jiang L, Jianxiong L, et al. A policy-guided imitation approach for offline reinforcement learning[J]. Advances in Neural Information Processing Systems, 2022, 35: 4085-4098.

[4] Garg D, Hejna J, Geist M, et al. Extreme q-learning: Maxent RL without entropy[J]. arXiv preprint arXiv:2301.02328, 2023.

[5] Xiao C, Wang H, Pan Y, et al. The in-sample softmax for offline reinforcement learning[J]. arXiv preprint arXiv:2302.14372, 2023.

[6] Zhang H, Mao Y, Wang B, et al. In-sample Actor Critic for Offline Reinforcement Learning[C]//The Eleventh International Conference on Learning Representations. 2022.

[7] Xu H, Jiang L, Li J, et al. Offline rl with no ood actions: In-sample learning via implicit value regularization[J]. arXiv preprint arXiv:2303.15810, 2023.

[8] Hansen-Estruch P, Kostrikov I, Janner M, et al. Idql: Implicit q-learning as an actor-critic method with diffusion policies[J]. arXiv preprint arXiv:2304.10573, 2023.

---

> ### Author Rebuttal · Authors · 2023-08-08
>
> We thank the reviewer for their efforts and time reviewing our paper.
>
> **Response to Weakness 1**: Thank you for your comment. **We want to emphasize that the original intention of CPED is not to directly apply the Flow-GAN idea to offline RL problem** for the following reasons.
>
> 1. **Advantage of Flow-GAN**: One key reason for selecting flow-GAN comes from its advantages. A clear advantage of Flow-GAN is its capability to provide an explicit and direct density estimator for the behavior policy. In contrast, other generative models (VAE) can only approximate the lower bound behavior density through ELBO.
>
> 2. **Further motivation for Flow-GAN**: We choose to use Flow-GAN as a density estimator because, first, we theoretically analyze in section 3.2 that estimating the density of the behavior policy in RL is equivalent to training a GAN. However, traditional GANs (including VAE) are essentially sample generators. To realize the idea of estimating a density function, we combine the Flow model with GAN to estimate the behavior density.
>
> 3. **Modifications to traditional Flow-GAN**: The Flow-GAN is originally designed for image generation problems, in our specific implementation, we make numerous adjustments to the structure of Flow-GAN (ref[1]) , including converting many CNN into fully connected networks and simplified some residual networks, so that overall network structure can adapt to the tasks of offline RL.
>
> 4. **Other Generative Models in offline RL**: In recent offline RL study, generative models become popular for behavior modeling owing to their strong model capacity, such as BCQ (using VAE), SPOT(using VAE), ref[3](using diffusion model). However, it is inappropriate to claim these studies simply borrow the VAE/diffusion model into offline RL domains.
> \end{itemize}
>
> With respect to the contribution of our manuscript, limited by the length of rebuttal, please find the bullet point **1.1** & **1.2** & **1.3** in “***Author Rebuttal by Authors***” part at the top of this page.
>
> Thus, we do not agree with the claim “*CPED borrow the idea of flow-gan into offline RL domains*”, and we believe the contribution of our CPED is underestimated.
>
> **Response to Weakness 2**:  In theory, we have demonstrated that the estimated density is accurate. Thus, as long as the density satisfies certain smoothness properties, we can prove that the estimated region is accurate. However, in practice, we are unable to observe all policies, making it challenging to quantify how well our estimation is. Nevertheless, the effectiveness of the experiments leads us to believe that the estimated policy is accurate, as the performance of offline RL heavily relies on the accuracy of the estimated region.
>
> **Response to Weakness 3**: Thank you for providing related references with diffusion technique. Actually, we have already cited [1] in the manuscript. We believe [1] is an important baseline for our CEPD. Compare [1] and CPED, we see that the CPED outperforms [1] for most D4RL and Antmaze tasks, which further shows the effectiveness of our CPED.
>
> **Response to Weakness 4**: We have provided further explanations about our contribution as well the comparison between flow-GAN and VAE (thus CEPD and SPOT) in both the response to Weakness 1 and “***Author Rebuttal by Authors***” part at the top of this page. Here we summarize the key points (limited by the rebuttal size, we do not list details for each point):
>
> 1.	**Similarities**: Please refer the "Similarities" part in Section 2 of “***Author Rebuttal by Authors***” part at the top of this webpage.
>
> 2.	**Advantage of CPED and Flow-GAN**: Please refer the part "Advantage of Flow-GAN" in **Response to Weakness 1** .
>
> 3.	**Deficiency of SPOT**: Please refer the "Deficiency of VAE" part in Section 2 of “***Author Rebuttal by Authors***” part at the top of this page.
>
> 4.	**Adaption of Flow-GAN and CPED for Estimating Behavior Policy**: Please refer to Further motivation for Flow-GAN in "**Response to Weakness 1**" above.
>
> 5.	**Modifications to traditional Flow-GAN** : Please refer to Modifications to traditional Flow-GAN in "**Response to Weakness 1** "above.
>
> Therefore, it is convincing that our proposed CPED has a clear edge against SPOT in both theory and experiments.
>
> **Response to Question 1**: Please refer our responses for Weakness 1 and Weakness 4 for our motivation for using CPED (flow-GAN) and more comparisons between CPED and SPOT (flow-GAN vs VAE).
>
> **Response to Question 2**:  We are afraid you may mis-understand our idea for optimizing Equation 7 in the manuscript. The actual objective function of Equation 7 is:
>
> $$\begin{aligned}
>  & \max_{\psi} \mathbb{E}_{s \sim \mathcal{P}\_\mathcal{D} ,\\, a \sim \pi\_{\psi}(\cdot|s), (s, a) \in  \tilde{\mathcal{S}} \times \tilde{\mathcal{A}} }[Q\_\eta(s,a)] \\\\
> & s.t.    \tilde{\mathcal{S}} \times \tilde{\mathcal{A}}= \\{(s,a) \in \mathcal{S} \times \mathcal{A}:  -\log L\_{\theta}^{\pi\_\beta}(s,a) < \epsilon \\}
> \end{aligned}\tag{A}$$
>
> which is a constrained optimization problem, and we are indeed using Lagrangian techniques to solve the constraint problem (A).
> For the constraint in problem (A), it is not hard constraints that exclude bad transitions as you mention. Instead it works as penalty terms in the Lagrange function (with Lagrangian multiplier $\alpha$) . In case other readers mis-understand our idea, in the revised manuscript, we will rewrite Equation 7 as Equation A under the form of constraint optimization problem.
>
> **Response to Lack of related work**: Thank you for your comment. We have already cited [1] and [2] in the manuscript. We appreciate you pointing out the research direction of in-sample learning paradigm, we will further cite [3-8] in the revised manuscript.
>
> ref[1]: https://arxiv.org/abs/1705.08868

---

> > ### Comment · Reviewer_1U1v · 2023-08-18
> > **Official Response by Reviewer 1U1v**
> >
> > Thank you for the clarifications provided, as most of they have addressed most of my concerns. I now understand the rationale behind using the Flow-GAN to estimate the density of the behavior policy. Specifically, the \textbf{further motivation for Flow-GAN} and Chapter 3.2 in the manuscript have become more convincing to me instead of finding a higher-fidelity density estimator. I also appreciate the authors' efforts to present more persuasive experiments demonstrating the efficiency of the Flow-GAN model by including additional comparisons with 'SfBC' in response to reviewer 6qDS. As a result, I have decided to raise my score.
> >
> > Furthermore, if the authors are willing to provide more compelling evidence as to why the Flow-GAN model outperforms the VAE model for estimating the behavior policy density, by conducting didactic toy examples, I would be inclined to further increase my rating.
> >
> > Other Questions: Is the code available?

---

> > > ### Author Response · Authors · 2023-08-20
> > > **(1/2) Response to Reviewer 1U1v**
> > >
> > > Thank you for your reply, and we appreciate your kind response.
> > >
> > > **Furthermore, if the authors are willing to provide more compelling evidence as to why the Flow-GAN model outperforms the VAE model for estimating the behavior policy density, by conducting didactic toy examples, I would be inclined to further increase my rating.**
> > >
> > > **Response:** Thank you for your comment. Of course we are glad to conduct additional studies comparing the VAE and flow-GAN as well as the advantages of CPED.
> > >
> > > Here, we firstly want to cite the Figure 1 in [1], which is a typical motivating example showing GAN-based model is much better than VAE-based model for approximating behavior policy.
> > >
> > > We are also conducting additional toy example comparing VAE and flow-GAN in generating behavior samples. Since we cannot attach further figures on openreview website at this stage, we will compare the likelihood of generated samples (by VAE and flow-GAN). We are still running the program, and we will soon provide the results in 1 day.
> > >
> > >
> > > In addition, in our response to 6qDs, we further compare BEAR, SPOT and CPED under the Halfcheetah-medium task, and the result figure is in our github code page <https://github.com/rl-study-group/rl_study_cped/tree/main> (github: Readme -> Note. Not allowed to attach further figures now). We can see that SPOT could reach a score of 6500 very fast, while CPED could achieve higher scores. both SPOT and CEPD significantly outperform BEAR.
> > >
> > >
> > > **Other Questions: Is the code available?**
> > >
> > > **Response:** Yes, we have uploaded the code at <https://github.com/rl-study-group/rl_study_cped/tree/main>.
> > >
> > > Note:
> > >
> > > * Please kindly search rl-study-group or rl\_study\_cped on <https://github.com/> if the above link is invalid due to the markdown grammar issue.
> > >
> > > * Due to the tight timeline for organizing the CPED code and uploading to github, some test code hasn't been completely removed. We will make an effort to tidy up all the code as soon as possible.
> > >
> > >
> > > [1] A Behavior Regularized Implicit Policy for Offline
> > > Reinforcement Learning: https://arxiv.org/pdf/2202.09673.pdf

---

> > > ### Author Response · Authors · 2023-08-21
> > > **(2/2) Follow up to Reviewer 1U1v**
> > >
> > > We have completed the toy-example experiments comparing VAE and Flow-GAN. Considering no further figures are allowed to be attached on openreview website at this stage, we compare the **mean of log-likelihood** of the generated samples in the following table. (We will attach the figure of generated samples in the revised manuscript)
> > >
> > >
> > > |Setting|  Ground Truth   | VAE Model | Flow-GAN  |
> > > |----|  ----  | ----  | ----  |
> > > |Setting 1| -2.89  | -140.83 |  -24.18
> > > |Setting 2| -2.78  | -50.05 |  -6.12
> > >
> > >
> > > Referring to Figure 1 in [1], it seems VAE-based model is not able to learn multi-modal data well. Thus, we select two simpler settings in our toy example.
> > >
> > > * **Setting 1**: We gather approximately $12,800,000$ data points from a multivariate normal distribution with a mean of $[1,9]$ and covariance matrix of $\Sigma = [[1,0],[0,1]]$.
> > >
> > > * **Setting 2**: We gather similar amount (with setting 1) data points from a mixture gaussian distribution, in which data are randomly drawn from two independent gaussian distributions ($\mu_1 =[1,1]$ and $\mu_2 = [9,9]$, $\Sigma_1 = \Sigma_2 = [[1,0],[0,1]]$) with equal probability.
> > >
> > > We then employ both VAE and flow-GAN to generate $1000$ samples and calculate the mean of log-likelihood. The table demonstrates that Flow-GAN outperforms VAE significantly in approximating the original data distribution.
> > >
> > >
> > > [1] A Behavior Regularized Implicit Policy for Offline Reinforcement Learning: https://arxiv.org/pdf/2202.09673.pdf

---

> ### Author Response · Authors · 2023-08-15
> **Look forward to your response**
>
> Thank you for taking the time to review our paper. We trust that our responses have effectively addressed the concerns you expressed in your review. Nevertheless, should you still have any lingering questions or unresolved matters, please feel free to inform us. We are committed to offering additional clarification and resolving any remaining issues to the best of our ability.

---

### Official Review · Reviewer_6qDS · 2023-07-06

**Soundness:** 3 good
**Presentation:** 2 fair
**Contribution:** 2 fair
**Rating:** 5
**Confidence:** 3

**Summary:**

This paper proposes a new offline RL algorithm called Constrained Policy optimization with Explicit Behavior density (CPED). The main idea is utilizing a flow-GAN model to estimate the probability density explicitly, which limits the exploration in the safe region. To demonstrate the effectiveness, the author provides some theoretical proofs for the convergence of CPED, and evaluates CPED in various standard offline RL tasks.

**Strengths:**

This paper is overall well-written, and it performs a comprehensive survey in offline RL and related fields such as inverse RL and generative model. The proposed method is well-motivated, which introduces flow-GAN to offline RL appropriately. This paper provides detailed theoretical analyses of the convergence to confirm the effectiveness of CPED.


**Weaknesses:**

1.	The novelty of this paper is somewhat limited as it primarily replaces the VAE model used in SPOT [1] with flow-GAN.
2.	With the exception of the ho-med dataset, CPED does not demonstrate clear improvements and, performs even worse than some previous methods in the experimental datasets.
3.	The implementation of CPED is not clearly stated, and this aspect will require further discussion in the questions.
4.	The usage of certain notations is confusing, and there are some grammar errors.


**Questions:**

1.	In Equation 7, the Lagrangian multiplier $\alpha$ is mentioned in the algorithm but cannot be found. Could you clarify the actual objective function? Additionally, the hyperparameter $\epsilon$ is not mentioned in your algorithm. How does $\epsilon$ affect the performance of your algorithm?
2.	In flow-GAN, specific neural networks are utilized for the generator and discriminator. Does this limitation affect their representational ability? Furthermore, does it result in a significant gap between the estimated policy and the behavioral policy?
3.	Previous work often considers online fine-tuning tasks after offline RL. Can CPED be applied to such tasks? If so, how does CPED compare to other offline RL methods in these scenarios?
4.	Figure 2 shows the learning curve comparison between BEAR and CPED. However, BEAR is considered somewhat outdated. As SPOT is a more recent offline RL method and seems more relevant to CPED, I'm particularly interested in the comparison between SPOT and CPED.
5.	In Section 5.2, you mention an additional trick using time-varying $\alpha$. It appears that this trick could be applicable to SPOT as well. Since the empirical improvement is limited, I'm curious if the improvement primarily results from this trick rather than flow-GAN. Could you provide additional results of SPOT with the time-varying $\alpha"?
6.	The choice of $\epsilon$ is not clearly explained. What is the specific quantile of the behavior policy density that you refer to? Does it correspond to the median or other quantiles? It would be helpful if you could provide examples of this "quantile" in several datasets.


**Limitations:**

Please refer to the weakness parts.

---

> ### Author Rebuttal · Authors · 2023-08-08
>
> Thank you for your review. Your comments are very useful for refining our manuscript.
>
> **Response to Weakness 1**:Thank you very much for this important question. **We want to emphasize that the original intention of CPED is not to directly apply the Flow-GAN idea to offline RL problem or to simply replace the method for estimating behavior policy in SPOT with GAN**.  We have clearly explained the original contribution of our CPED method and as well the motivation for using Flow-GAN in the “***Author Rebuttal by Authors***” part at the top of this page.
>
> For the contribution of our paper, please refer the bullet point **1.1 & 1.2 & 1.3 in Section 1** in  “***Author Rebuttal by Authors***” part at the top of this page
>
> For the issue why we choose flow-GAN instead of other methods: please refer **Similarities/Advantage of Flow-GAN/Deficiency of VAE model/Further motivation for Flow-GAN in Section 2** in  “***Author Rebuttal by Authors***” part at the top of this page.
>
> **Response to Weakness 2**: In the manuscript, the selected competitors (DT/SPOT/IQL) are among the SOTA methods for D4RL tasks, resulting in high baselines. From Table 1&2 in our manuscript, our CPED achieves the best performance for hop-med task, and provides competitive performances with DT and SPOT for hop-med-exp and hop-med-rep tasks. For other tasks such as half-cheetah and walk2d,  CPED delivers the best performances in 4 out of 6 tasks.
>
> In addition, the total score of the CPED is significantly higher than other competitors, thus it is clear that the proposed CPED performs better than the competitors, and it also contributes to SOTA methods for offline D4RL tasks.
>
> **Response to Weakness 3 & 4**: We will further refine the manuscript, especially revising notations and the implementation part (Algorithm 1, Section 5 and Appendix E),  so that the manuscript is more readable. For further discussion in Questions, please refer to Response to Questions later.
>
> **Response to Question 1**:  We did not specifically mention $\alpha$ in Equation 7 because of the limited page. In fact, the problem of equation 7 is a constraint optimization problem, and the actual objective function is:
> $$\begin{aligned} & \max_{\psi} \mathbb{E}_{s \sim \mathcal{P}\_\mathcal{D} ,\\, a \sim \pi\_{\psi}(\cdot|s), (s, a) \in  \tilde{\mathcal{S}} \times \tilde{\mathcal{A}} }[Q\_\eta(s,a)] \\\\
> & s.t.    \tilde{\mathcal{S}} \times \tilde{\mathcal{A}}= \\{(s,a) \in \mathcal{S} \times \mathcal{A}:  -\log L\_{\theta}^{\pi\_\beta}(s,a) < \epsilon \\}
> \end{aligned}\tag{A}$$
>
> We believe a natural solution for the constraint problem (A) is to apply Lagragian method. Thus, we did not specifically mention  Lagragian multiplier $\alpha$ in Equation 7 for simplicity (only show in Algorithm 1). In case other readers mis-understand our idea, we will rewrite Equation 7 as Problem (A) in the manuscript, and emphasize $\alpha$ is the Lagragian multiplier.
>
> With respect to the effect of $\epsilon$, limited by the page, the ablation study of $\epsilon$ is in Appendix E, in which we consider several choices of $epsilon$ in various D4RL tasks.
>
> **Response to Question 2**: The specific neural network in flow-GAN comes from the flow model. In the original paper of Flow-GAN [1], the power and representation ability of flow-GAN are fully shown from Table 1&Table 2 in [1] in image application. In other references[2,3], even the generator is assumed to be invertible, it has not significantly impacted the performance of Flow-GAN [2,3]. The representation power of Flow-GAN remains highly competitive and effective despite this assumption.
>
> **Response to Question 3**: We really appreciate your useful comment. Since the focus of our manuscript is offline reinforcement learning, online fine-tuning is not a primary topic for us. Considering some offline RL methods (SPOT/IQL) treat online fine-tuning as extensions of their papers, the proposed CPED is also capable of downstream online training and we would like to leave this topic as future work.
>
> Following Section 5.3 in SPOT paper, the performance of each method (SPOT/IQL) is further improved after online fine-tuning, and the SPOT performs better than IQL. Since CPED is less conservative than both IQL and SPOT, and the offline performance of CPED is significantly better than SPOT, it is expected that the performance of CPED after online fine-tuning is better than both IQL and SPOT.
>
> **Response to Question 4**: The main purpose of Figure 2 is to demonstrate that our CPED offers more flexibility compared to traditional distance-based control methods, and it overcomes the issue of being too conservative. Consequently, from the learning curve, the CPED provides a broader "safe area," leading to an upward trend in average return, while BEAR, due to its conservatism, does not show significant performance improvement. We believe that SPOT would likely demonstrate similar outcomes as it also tends to be less conservative.
>
> **Response to Question 5**:  We perform extra experiments for SPOT with time varying $\alpha$ for Antmaze and two D4RL tasks, and the resulting figure is shown in the **pdf file** in ***Author Rebuttal by Authors***. We can see that the performances with time varying $\alpha$ are close or even inferior to those with constant $\alpha$, indicating the trick of varying $\alpha$ does not result too much benefit. Therefore, we can conclude the improvement indeed come from the Flow-GAN model, rather than the tricks in model training.
>
> **Response to Question 6**: We believe you may misunderstand our idea in setting $\epsilon$. We are not using the quantile of behavior policy density as $\epsilon$ (used in other methods such as SPOT), instead we use the mean likelihood of the training batch in the experiment.
>
> The choice of $\epsilon$ is illustrated in Section 5.2, Line 324-327: “A commonly used setting is ...”
>
> [1]https://arxiv.org/abs/1705.08868
>
> [2]https://arxiv.org/abs/1410.8516
>
> [3]https://arxiv.org/abs/1605.08803

---

> > ### Comment · Reviewer_6qDS · 2023-08-18
> >
> > Thank you for the clarification and additional experimental results. The rebuttal has addressed most of my concerns. However, the reviewer still feels it necessary to perform more detailed experiments to compare CPED with SPOT to confirm the superiority of CPED since SPOT is **fairly relevant** to CPED. Besides, the performance of SPOT is much better than BEAR, as shown in Table 2. The authors should not claim “SPOT would likely demonstrate similar outcomes” without further experiments. Besides, the reviewer is curious about the meaning of d4rl score in the pdf file and suggests that the same metrics (average return) should be used here to make it convenient to compare the results.

---

> > > ### Author Response · Authors · 2023-08-20
> > > **(1/2) Response to Reviewer 6qDS**
> > >
> > > Thank you for your response. We are sorry for some unclear statements and definitions in our last response. We have added extra experiements, and corrected unclear statements and definitions in the following.
> > >
> > > **However, the reviewer still feels it necessary to perform more detailed experiments to compare CPED with SPOT to confirm the superiority of CPED since SPOT is fairly relevant to CPED.**
> > >
> > > **Response:** Thank you for your comment. In last 2 days, we add extra experiments further comparing SPOT and CPED. Per the request of reviewer 1U1v, we have released our code on github  <https://github.com/rl-study-group/rl_study_cped/tree/main>, where we also add the addtional comparison figure (github: Readme -> Note. We are currently not allowed to attach further figures on openreview website at this stage. Limited by time, we only finish the half-cheetah task for demonstration, and we will add more task results in the revised manuscript). Please note as we haven't disclosed any personal information in this github link, this **does not** violate the double-blind reviewing policy.
> > >
> > > From the figure in the link, we can see that the average return of SPOT could quickly reach a score of 6500 and then fluctuates around this level. Neverthesless, although CEPD is not as fast as SPOT, it could finally reach a higher score around 7000.
> > >
> > > **Besides, the performance of SPOT is much better than BEAR,as shown in Table 2. The authors should not claim ''SPOT would likely demonstrate similar outcomes'' without further experiments.**
> > >
> > >
> > > **Response:** We apologize for the unclear statements. In our last response, we want to claim **''SPOT would likely demonstrate similar outcomes''**  ***with CPED, not with BEAR***, since both SPOT and CPED are less conservative. As shown in the link above, we have compared BEAR, SPOT and CPED, and we can see that both SPOT and CEPD significantly outperform BEAR, and CPED works better than SPOT.
> > >
> > >
> > > **Besides,the reviewer is curious about the meaning of d4rl score in the pdf file and suggests that the same metrics (average return) should be used here to make it convenient to compare the results.**
> > >
> > > **Response:** Thank you for pointing out our unclear definition. Actually, the definition of *d4RL Score* in our manscript (defined from SPOT source code) is **d4rl\_score = normalized_score/100**, where *normalized_score* is **the standard definition for d4rl tasks** from the original d4rl paper[1]. In the SPOT paper [2], the authors also call *normalized_score* as *normalized return*, and the name **d4rl\_score** comes from the  results of SPOT after running the source code.
> > >
> > > Since we cannot attach further figure here, we will revise the metric as **''normalized score''** for consistency (same metric with the original d4rl paper, SPOT and other references) in the revised manuscript. Actually the shape of the figure remains unchanged, only the vertical axis will be scaled up by a factor of 100.
> > >
> > > [1] d4rl paper: https://arxiv.org/pdf/2004.07219.pdf.
> > >
> > > [2] SPOT: https://openreview.net/pdf?id=KCXQ5HoM-fy

---

> > > ### Author Response · Authors · 2023-08-21
> > > **(2/2)Follow up to Reviewer 6qDS**
> > >
> > > In this response, we provide additional evidence by presenting the outcomes of new experiments comparing the VAE and Flow-GAN models, as per the comment from Reviewer 1U1v.
> > >
> > > Firstly, we intend to cite Figure 1 in [1], a nice illustrative example, to show that GAN-based models outperform VAE-based models in approximating behavior policies.
> > >
> > > Furthermore, we conduct additional toy example experiments by comparing the performance of VAE and flow-GAN in generating behavior samples. Due to constraints on attaching additional figures to the OpenReview website at this stage, we present a comparison of the **mean of log-likelihood** of generated samples in the table below. (We will attach the figure of generated samples in the revised manuscript)
> > >
> > >
> > > |Setting|  Ground Truth   | VAE  Model | Flow-GAN  |
> > > |----|  ----  | ----  | ----  |
> > > |Setting 1| -2.89  | -140.83 |  -24.18
> > > |Setting 2| -2.78  | -50.05 |  -6.12
> > >
> > > Referring to Figure 1 in [1], it seems VAE-based model is not able to learn multi-modal data well. Thus, we select two simpler settings in our toy example.
> > >
> > > * **Setting 1**: We gather approximately $12,800,000$ data points from a multivariate normal distribution with a mean of $[1,9]$ and covariance matrix of $\Sigma = [[1,0],[0,1]]$.
> > >
> > > * **Setting 2**: We gather similar amount (with setting 1) data points from a mixture gaussian distribution, in which data are randomly drawn from two independent gaussian distributions ($\mu_1 =[1,1]$ and $\mu_2 = [9,9]$, $\Sigma_1 = \Sigma_2 = [[1,0],[0,1]]$) with equal probability.
> > >
> > > We then employ both VAE and flow-GAN to generate $1000$ samples and calculate the mean of log-likelihood. The table demonstrates that Flow-GAN outperforms VAE significantly in approximating the original data distribution.
> > >
> > >
> > > [1] A Behavior Regularized Implicit Policy for Offline Reinforcement Learning: https://arxiv.org/pdf/2202.09673.pdf

---

> ### Author Response · Authors · 2023-08-15
> **Look forward to your response**
>
> Thank you so much for reviewing our paper, and we appreciate your helpful suggestions.  We sincerely hope that our responses have adequately addressed the concerns you raised in your review. However, if you still have any unresolved concerns or additional questions, please do not hesitate to let us know. We would be more than happy to provide further clarification and address any remaining issues.

---

### Official Review · Reviewer_Y4vE · 2023-07-06

**Soundness:** 3 good
**Presentation:** 2 fair
**Contribution:** 2 fair
**Rating:** 6
**Confidence:** 3

**Summary:**

This work proposes to use flow-GAN model to explicitly model the distribution of the behaviour policy and use that to alleviate the OOD actions problem in offline RL.
A theoretical analysis of convergence is presented. Empirical evaluation on d4rl gym locomotion and antmaze show that the proposed CPED algorithm is very competitive.

**Strengths:**

* The method is novel and very intuitive. Instead of using all different tricks to avoid OOD, explicitly modelling the behaviour policy is much more straightforward.
* The writing is clear
* Empirical evaluation shows the performance of CPED is comparable to SOTAs

**Weaknesses:**

* Compared to prior methods like IQL, DT, or even CQL. The proposed method has an additional generative modelling module, which makes the overall architecture more complex.
* Again, the empirical results are good but it's technically not surpassing SOTA. For gym-locomotion, there are algorithms like ATAC [1] and SAC-N [2] that have similar level of performance. As for antmaze, the performance of CPED is inferior than IQL, which is not the SOTA for offline RL in general. For model-based methods, there're methods like trajectory transformer [3] that perform better than IQL.

[1] https://proceedings.mlr.press/v162/cheng22b/cheng22b.pdf

[2] https://arxiv.org/pdf/2110.01548.pdf

[3] https://arxiv.org/abs/2106.02039

Minor issues:

> CPED can accurately identify the safe region and enable exploration within the region

I would avoid using the term "exploration" here because it can hint to readers that the proposed method can be helpful for online learning but I believe this is not what the authors mean here.

**Questions:**

* What's the computational overhead of training this Flow-GAN?
* Why do you pick Flow-GAN? Why not use simpler ones like VAE, or even just discretization?

**Limitations:**

Yes

---

> ### Author Rebuttal · Authors · 2023-08-08
>
> Thank you so much for your review. We appreciate your valuable questions and suggestions. We have replied your comments in detail, and please see the following for your response.
>
> **Response to Weakness 1**:  We agree that the proposed CPED introduces an additional generative modeling module in the offline RL solution. Nevertheless, this will not lead to an over complex architecture.
>
> In recent studies of offline RL methods, many generative modeling modules (VAE/GAN/diffusion model) are suggested for modeling behavior policies (BCQ/SPOT), and the sizes of such modules does not pose significant issues. For CPED, the training cost of flow-GAN is close to that of GAN model, and both models are much more efficient than diffusion models. Thus, the complexity of CPED is acceptable. In addition, it is noted that the size of DT will be significantly large when the sequence/input trajectory is long.
>
> **Response to Weakness 2**:  Thank you for pointing out the related references and methods. We will certainty include the above references and related methods in the revised manuscript (we have cited [2] already). According to references [1] and [2], both ATAC and SAC-N provide competitive performances with CPED on hopper and halfcheetah tasks, while CPED performs slightly better on walker2d tasks.
>
> We would like to clarify that the SAC-N method utilizes multiple Q-functions, which may contribute to better performances but can also lead to computational issues. In contrast, CPED employs only 2 Q-networks. In addition, the trajectory transformer methods lack strong theoretical interpretability and it requires extra parameters such as “return-to-go”. Therefore, we believe that the proposed CPED has potential for further improvement when multiple Q-functions or other modeling tricks are applied.
> As we mentioned in the “author rebuttal” in the top, our CPED does not solely aim to be SOTA in experiments. Moreover, we want to show CPED has great potential in further improvement, and we will continue to delve into this aspect in the future.
>
> **Response to Minor issues**:  In response to your feedback, we will make further revisions to the term "exploration" to ensure that readers are not misled. The sentence will be revised as “CPED can accurately identify the safe region and allow policy learning within the region”
>
> **Response to Question 1**:  From our experiment, training the flow-GAN will indeed incurs additional computational overhead. Nevertheless, this computational overhead is not a significant concern for the overall training process. In our study, the training time of CPED is only approximately 10% longer compared with the traditional offline RL methods such as BEAR and BRAC. Therefore, the training cost of CPED is quite acceptable.
>
> **Response to Question 2**: Thank you very much for this important question. We have clearly explained the advantage as well the motivation for using Flow-GAN in the “author rebuttal” part. We are glad to re-illustrate the points here so that you can understand our motivation better.
>
> For the issue why we choose flow-GAN instead of other generative methods (e.g. VAE):
>
> 1.	**Similarities**: In the field of policy control methods for offline RL problem, the idea of ensuring the consistency of the support of learned policy and that of the behavior policy is considered the most desirable approach to tackle distribution shift. Both CPED and SPOT aim to achieve this idea, but they do it differently.
>
> 2.	**Advantage of Flow-GAN**: A clear advantage of Flow-GAN is its capability to provide an explicit and direct density estimator for the behavior policy. In contrast, VAE can only approximate the lower bound behavior density through ELBO.
>
> 3.	**Deficiency of VAE model**: Another experimental evidence comes from ref[1] and ref[2]. In ref[1], the author claims typically-used VAEs do not align well with the behavior dataset, making it challenging to effectively cover the behavior policy distribution. The author further presents an experiment result that the model performance is significantly reduced when the VAE architecture is used (Compared Fig2 to Fig 6 ). In addition, this claim is also validated in ref[2] (in appendix D), where the performance of the method with VAE structure is inferior to that of the original method.
>
> 4.	**Further motivation for Flow-GAN**: We chose to use Flow-GAN as a density estimator because, first, we theoretically analyze in section 3.2 that estimating the density of the behavior policy in RL is equivalent to training a GAN. However, traditional GANs are essentially sample generators. To realize the idea of estimating a density function, we combine the Flow model with GAN to estimate the behavior density. The Flow-GAN is originally designed for image generation problems, in our specific implementation, we made numerous adjustments to the structure of Flow-GAN (ref[3]) , including converting many CNN into fully connected networks and simplified some residual networks, so that overall network structure can adapt to the tasks of offline RL. We believe the flow-GAN model is promising, and hope that future work will further optimize the design of Flow-GAN's structure, allowing it to unleash its true potential on more challenging behavior policy datasets.
>
> 5.	**Discretization**: As for discretization, in the currently prevalent continuous control tasks, this method faces difficulties in accurately estimating the behavior policy density.
>
>
> ref[1]: https://arxiv.org/pdf/2007.11091.pdf
>
> ref[2]: https://arxiv.org/pdf/2209.14548.pdf
>
> ref[3]: https://arxiv.org/abs/1705.08868

---

> > ### Comment · Reviewer_Y4vE · 2023-08-14
> > **Response to the Rebuttal**
> >
> > I would thank the authors for their clarification. I like the fact that the proposed method has only 10% computational overhead.
> >
> > For the reason of defending Flow-GAN, the arguments are not strong enough to convince me, especially without further empirical results.
> >
> > I'll keep the score for now.

---

> > > ### Author Response · Authors · 2023-08-15
> > > **Response to the Reviewer & Thank you for you comments**
> > >
> > > **I would thank the authors for their clarification. I like the fact that the proposed method has only 10% computational overhead.**
> > >
> > > **Response:** Thank you so much for your nice comment.
> > >
> > >
> > > **For the reason of defending Flow-GAN, the arguments are not strong enough to convince me, especially without further empirical results.**
> > >
> > >
> > > **Response:** Thank you for your comment. To provide further empirical results and demonstrate the advantages of the flow-GAN, we conduct comparisons between CPED, SPOT[1], and SfBC[2]. In SPOT, the VAE model is employed to approximate the lower bound of the behavior likelihood (also compared in our manuscript). On the other hand, SfBC employs diffusion techniques, another type of generative model, to estimate the behavior model. The results of each method for D4RL tasks are presented in the table below.
> > >
> > >
> > > |  Task   | CPED  | SPOT   | SfBC  |
> > > |  ----  | ----  | ----  | ----  |
> > > | HalfCheetah-Medium  | **61.8** | 58.4  | 45.9 |
> > > | Hopper-Medium  | **100.1** | 86.0  | 57.1 |
> > > | Walker-Medium  | **90.2** | 86.4  | 77.9 |
> > > | HalfCheetah-Medium-replay  | **55.8** | 52.2  | 37.1 |
> > > | Hopper-Medium-replay  | 98.1 | **100.2** | 86.2 |
> > > | Walker-Medium-replay  | **91.9** | 91.6  | 65.1 |
> > > | HalfCheetah-Medium-expert  | 85.4 | 86.9  | **92.6** |
> > > | Hopper-Medium-expert | 95.3 |  99.3 | **108.6** |
> > > | Walker-Medium-expert  | **113.04** | 112.0  | 109.8 |
> > >
> > >
> > > From the table, it becomes evident that CPED outperforms its competitors across various tasks. Coupled with the reasoning analysis provided in our previous response, we are confident that Flow-GAN stands as a superior selection compared to alternative tools for behavior model estimation.
> > >
> > >
> > > [1] Supported policy optimization for offline reinforcement learning. NeurlIPS 2022.
> > >
> > > [2] Offline reinforcement learning via high-fidelity generative behavior modeling. ICLR 2023.

---

### Author Rebuttal · Authors · 2023-08-08

We are grateful for the valuable questions and suggestions were given by all four reviewers that help us to revise our manuscript. After reading all the reviews, we have answered each reviewer's questions in detail, and also added relevant experiments based on each reviewer's suggestions to support our work.

As some reviewers still have concerns on the contribution of our manuscript as well as selecting the Flow-GAN instead of other methods (e.g. VAE), **we want to emphasize that the original intention of CPED is not to directly apply the Flow-GAN idea to offline RL problem or to simply replace the method for estimating behavior policy in SPOT with GAN**. Furthermore, we want to illustrate following issues:

1. For the contribution of our manuscript, through our exploration of the CPED method, we aim to provide the following conclusions from both theoretical and experimental perspectives:
    - **1.1** The combination of GAN and MLE-based density estimation (e.g. Flow-GAN) is more suitable for estimating behavior policies in offline RL, both in theory and experiments. We provide a theoretical guarantee indicating that the CPED can access the optimal Q-function, and extensive experiments show that CPED substantially outperforms state-of-the-art methods.
    - **1.2** Keeping the support of the learned policy close to that of the behavior policy is the key idea in policy control methods. In our proposed CPED, by introducing an explicit density function, we can effectively achieve this objective.
    - **1.3** CPED is not only intended to demonstrate its superiority through experimental results but also aims to advance the completeness of policy control methods. In fact, we believe that the Flow-GAN density estimator has significant potential, considering its representation of the policy density learning process in theory and its promising performance in experiments. We also hope that our work can serve as a catalyst, driving the development of the idea of estimating behavior policies to address offline RL problems.

    The above three points are our fundamental contributions to the offline RL community.

2. For the issue why we choose flow-GAN instead of other generative methods (e.g. VAE):
    - **Similarities**: In the field of policy control methods for offline RL problem, the idea of ensuring the consistency of the support of learned policy and that of the behavior policy is considered the most desirable approach to tackle distribution shift. Both CPED and SPOT aim to achieve this idea, but they do it differently. CPED utilizes flow-GAN to estimate the density of behavior policy while SPOT selects the VAE as the tools.
    - **Advantage of Flow-GAN**: A clear advantage of Flow-GAN is its capability to provide an explicit and direct density estimator for the behavior policy. In contrast, VAE can only approximate the lower bound behavior density through ELBO.
    - **Deficiency of VAE model**: Another experimental evidence comes from ref[1] and ref[2]. In ref[1], the author claims typically-used VAEs do not align well with the behavior dataset, making it challenging to effectively cover the behavior policy distribution. The author in ref[1] further presents an experiment result that the model performance is significantly reduced when the VAE architecture is used (Compared Fig2 to Fig 6). In addition, this claim is also validated in ref[2] (in appendix D), where the performance of the method with VAE structure is inferior to that of the original method.
    - **Further motivation for Flow-GAN**: We choose Flow-GAN as a density estimator because, first, we theoretically analyze in section 3.2 that estimating the density of the behavior policy in RL is equivalent to training a GAN. However, traditional GANs (including VAE) are essentially sample generators. To realize the idea of estimating a density function, we combine the Flow model with GAN to estimate the density of behavior policy. The Flow-GAN is originally designed for image generation problems, in our specific implementation, we make numerous adjustments to the structure of Flow-GAN (ref[3]) so that it can adapt to the offline RL tasks. We hope that future work will further optimize the design of Flow-GAN's structure, allowing it to unleash its true potential on more challenging behavior policy datasets.

* ref[1]: https://arxiv.org/pdf/2007.11091.pdf
* ref[2]: https://arxiv.org/pdf/2209.14548.pdf
* ref[3]: https://arxiv.org/abs/1705.08868

---

### Decision · Program_Chairs · 2023-09-21

**Decision:**

Accept (poster)

**Comment:**

### Summary

The paper introduces an offline reinforcement learning (RL) method called Constrained Policy optimization with Explicit Behavior density (CPED) to address the challenge of estimating Out-of-Distribution (OOD) data points. Existing solutions for this problem either overly restrict policies or struggle to identify OOD regions accurately. CPED employs a flow-GAN model to explicitly estimate the density of behaviour policy, allowing it to precisely identify safe exploration regions and thereby achieve less conservative learning policies. The theoretical analysis demonstrates the convergence of CPED, and empirical evaluations on standard offline RL tasks, including d4rl gym locomotion and ant-maze, show that CPED is highly competitive, outperforming existing alternatives and yielding higher expected returns.


### Decision

The paper is well-written and easy to understand. The CPED algorithm technically reasonable and achieves significantly better results than other algorithms. The reviewers had some concerns regarding to the novelty of the algorithm concerned with other similar algorithms like SPOT. However, the authors did a good job addressing these concerns. The ideas in this paper is worth sharing with the rest of the NeurIPS community. I would recommend the authors to address some of the concerns raised by the reviewers in the camera-ready version of the paper.